# Immediate or Delayed Transplantation of a Vein Conduit Filled with Nasal Olfactory Stem Cells Improves Locomotion and Axogenesis in Rats after a Peroneal Nerve Loss of Substance

**DOI:** 10.3390/ijms21082670

**Published:** 2020-04-11

**Authors:** Maxime Bonnet, Gaëlle Guiraudie-Capraz, Tanguy Marqueste, Stéphane Garcia, Charlotte Jaloux, Patrick Decherchi, François Féron

**Affiliations:** 1Aix Marseille Univ, CNRS, ISM, UMR 7287, Institut des Sciences du Mouvement: Etienne-Jules MAREY, Equipe Plasticité des Systèmes Nerveux et Musculaire (PSNM), Parc Scientifique et Technologique de Luminy, Faculté des Sciences du Sport de Marseille, CEDEX 09, F-13288 Marseille, France; maxime.bonnet@univ-amu.fr (M.B.); tanguy.marqueste@univ-amu.fr (T.M.); patrick.decherchi@univ-amu.fr (P.D.); 2Aix Marseille Univ, CNRS, INP, UMR 7051, Institut de Neuropathophysiologie, Equipe Nasal Olfactory Stemness and Epigenesis (NOSE), CEDEX 15, F-13344 Marseille, France; gaelle.guiraudie@univ-amu.fr (G.G.-C.); charlotte.jaloux@ap-hm.fr (C.J.); 3APHM, Laboratoire d’Anatomie Pathologique, Hôpital Nord, Chemin des Bourrely, CEDEX 20, F-13915 Marseille, France; stephane.garcia@ap-hm.fr; 4APHM, Unité de Culture et Thérapie Cellulaire, Hôpital de la Conception, F-13006 Marseille, France

**Keywords:** nerve repair, neglected wound, biomaterial, stem cells, syngenic transplantation, ecto-mesenchymal, electrophysiology, neurofilament, myelin

## Abstract

Over the recent years, several methods have been experienced to repair injured peripheral nerves. Among investigated strategies, the use of natural or synthetic conduits was validated for clinical application. In this study, we assessed the therapeutic potential of vein guides, transplanted immediately or two weeks after a peroneal nerve injury and filled with olfactory ecto-mesenchymal stem cells (OEMSC). Rats were randomly allocated to five groups. A3 mm peroneal nerve loss was bridged, acutely or chronically, with a 1 cm long femoral vein and with/without OEMSCs. These four groups were compared to unoperated rats (Control group). OEMSCs were purified from male olfactory mucosae and grafted into female hosts. Three months after surgery, nerve repair was analyzed by measuring locomotor function, mechanical muscle properties, muscle mass, axon number, and myelination. We observed that stem cells significantly (i) increased locomotor recovery, (ii) partially maintained the contractile phenotype of the target muscle, and (iii) augmented the number of growing axons. OEMSCs remained in the nerve and did not migrate in other organs. These results open the way for a phase I/IIa clinical trial based on the autologous engraftment of OEMSCs in patients with a nerve injury, especially those with neglected wounds.

## 1. Introduction

Peripheral nerve injuries account for 2.8% of trauma [1] and are usually responsible for a partial loss of motor and sensitive function or even paralysis. The most frequently injured nerves are the ulnar, radial, median, peroneal, and sciatic nerves [1,2]. Strategies to improve peripheral nerve regeneration are clinically important. Injured nerves do not spontaneously recover their function when axoplasmic flow is interrupted and, very often, nerve continuity must be restored by a microsurgical procedure. Currently, epineural repair is the most appropriate surgical procedure when the nerve is transected and can be sutured without generating an undue tension [3]. However, peripheral nerve injuries resulting in large gaps cannot be simply end-to-end sutured without generating a tension that impedes axonal regeneration.

When nerve endings cannot be connected without tension, autografts are used for nerve reconstruction. The clinician removes a section of the sural or other sensory nerve and inserts it between the two nerve stumps. Nonetheless, autologous nerve graft is associated with donor-site morbidity, including neuroma formation, scarring and permanent loss of function. Furthermore, the donor nerves are not necessarily adapted to the receiver nerve caliber and they are in limited numbers. Alternatively, vein conduits can be used. Described for the first time in 1909 by Wrede [4] and, later on, by Chiu et al. (1982) [5], they overcome some of the above-mentioned limitations. Vein bridges are abundant, easily accessible, available in a multitude of 3D sizes and free of downstream effects. No additional lesion and no inflammation are associated to this procedure. Overall, a venous conduit is a suitable clinical material since it is biologically inert, inexpensive, biocompatible, thin, flexible, and transparent [6]. Previous studies indicate that this bridging strategy is beneficial, leading to partial sensory-motor [7,8,9], electrophysiological and histological recoveries [10,11,12]. For smaller gaps (inferior to 2 cm), nerve conduit and nerve autograft seem to have the same efficacy on sensory-motor recovery [13]. In addition, when the segment defect is long enough to possibly induce a collapse of the bridge, the inserted vein can be filled with fresh skeletal muscle (for a review, [14]).

Besides, cell therapy can be performed as an additional approach. Human stem cells stand as a material of choice for scientists who wish to implement regenerative medicine [15]. However, to achieve these exciting promises, important choices have to be made, the first of which being the selection of the most appropriate stem cell subtype. Among the top candidates for brain research, we can cite the human nasal olfactory mucosa, home of a permanent neurogenesis [16], which harbors multipotent stem cells [17,18], belonging to the ecto-mesenchymal stem cell family [19], and also ensheathing cells that have been assessed for repairing peripheral nerves (for a review, [20]).

This neural crest-derived tissue remains in a state of embryo-like development and is used as well as its associated olfactory ecto-mesenchymal stem cells (OEMSCs) for repairing the pathological or traumatized brain. Endowed with specific immuno-modulatory properties [21] and easily accessible, under local anesthetic, in every living adult individual [22], OEMSCs can be used for autologous as well as allogenic transplantation. Interestingly, they can even cross various metabolic barriers [23,24], and therefore be injected in the cerebrospinal fluid or the blood circulation. Promising results were obtained, with various modes of grafting, in animal models of paraplegia [25,26], cochlear damage [27,28], amnesia [24], or peripheral nerve injury [29].

In order to further assess the therapeutic potential of OEMSCs, we designed a study in which a 3 mm rat peroneal nerve segment was removed and a venous conduit was sutured on each nerve stump, either immediately or two weeks after the trauma. Syngeneic OEMSCs were combined with thrombin and fibrinogen and inserted or not into the venous bridge, two weeks after the trauma. Transplanted and non-transplanted rats were compared to unoperated animals. The chronic model mirrors the delayed repair associated to neglected wounds in human care and, in line with a clinical trial based on autologous stem cell transplantation, the time required for cultivating, in GMP conditions, about 10 million human OEMSCs.

## 2. Results

No clinical sign of pain or unpleasant sensation (i.e., auto-mutilation, screech, prostration, hyperactivity, anorexia) and no paw-eating behavior were observed during the 12 week-long study.

### 2.1. OEMSCs Improves Locomotion but Not Muscle Weight

The olfactory mucosa being easily accessible in every human, autologous transplants can be performed in clinics. However, such a procedure, although not impossible, is difficult to implement in rodents. This is the reason why we opted for syngeneic rats. These rodents are almost genetically identical and immunologically compatible, allowing transplantation without the use of immunosuppressant. As shown on Appendix A, cultivated cells were positive for nestin, CD73, CD90 and negative for CD45, indicating that they belong to the OEMSC family and not the hematopoietic stem cell lineage.

Figure 1A summarizes the locomotor behavior of rats with an immediate insertion of a vein conduit after a loss of nerve substance. Overall, the kinematics between the two groups with a vein transplant (immediate vein graft (IVG) and immediate vein graft and stem cells (IVG-SC)) only varies during the first six weeks and, three months after the trauma, animals from both groups display a similar locomotion. Figure 1B compares the locomotor recovery of rats with a delayed insertion of a vein bridge. Throughout the assessment period, the locomotion of animals without stem cells (DVG) is suboptimal. The PFI value is always lesser to the score of the IVG rats (Figure 1A,B). A similar trend is observed when comparing DVG-SC rats to IVG-SC rats: a statistically significant difference is observed at W4 (*p* < 0.001), W6 (*p* < 0.001), W8 (*p* < 0.01), W10 (*p* < 0.01). However, three months post-surgery, the PFI value of DVG-SC rats (−21.27 ± 1.57) is similar to the PFI value of IVG-SC rats (−18.05 ± 1.27). Overall, the functional recovery of DVG rats is always significantly reduced when compared to DVG-SC rats.

A nerve section induces a weight loss in the target muscle, namely the tibialis anterior. Figure 1C indicates that, 12 weeks post-surgery, the ratio “muscle weight/body weight” is significantly reduced in both groups with a vein bridge (IVG, IVG-SC), when compared to the Control group. However, no significant difference is observed between the two groups with a loss of nerve substance. A similar ratio is observed between the DVG and DVG-SC groups (Figure 1D). Nonetheless, a significant difference between the DVG-SC and IVG-SC groups (*p* < 0.01) is noticed, the IVG-SC group exhibiting a higher ratio.

### 2.2. OEMSCs Partially Maintain the Contractile Phenotype of the Target Muscle

Contractions of the muscle tibialis anterior were elicited by electrical stimulation of the peroneal nerve. Analysis of the MCR/A ratio indicates that the phenotype of the target muscle is modified in stem cell-free animals (IVG and DVG groups) indicating a shift to a slower phenotype. Conversely, the implanted stem cells allow the IVG-SC and DVG-SC animals to maintain a phenotype close to a control situation (Figure 2A,B). In addition, the DVG-SC animals display a significant improvement (*p* < 0.05) of the MRR/A ratio when compared to the ungrafted animals (DVG) (Figure 2D). No difference between IVG vs. DVG and IVG-SC vs. DVG-SC groups is observed.

### 2.3. OEMSCs Do Not Enhance Nerve Afferent Response

#### 2.3.1. Response to Electrically Induced Fatigue

Afferent responses are significantly (*p* < 0.05 and *p* < 0.01) lower in all vein-bridged animals, even when stem cells were grafted in the vein conduit (Figure 3A,B). No difference between IVG vs. DVG and IVG-SC vs. DVG-SC groups is observed.

#### 2.3.2. Response to Lactic Acid Injection

A significant lower response is observed in the IVG (*p* <0.05), DVG (*p* <0.01) and DVG-SC (*p* <0.01) groups, after injection of LA. Interestingly, no statistically significant difference is observed between the Control and the IVG-SC groups (Figure 3C,D). Of note, the grafting of stem cells in the delayed animal model (DVG-SC) induces a recovery similar to the non-delayed and ungrafted group (IVG). In addition, a significant difference is observed when comparing IVG and DVG groups (*p* < 0.05).

#### 2.3.3. Response to Potassium Chloride (KCl) Injection

A significant lower response is observed in the IVG (*p* < 0.05), IVG-SC (*p* < 0.01), DVG (*p* < 0.01) and DVG-SC (*p* < 0.01) groups after injection of KCl. Grafting of stem cells does not improve the recovery (Figure 3E,F). No difference between IVG vs. DVG and IVG-SC vs. DVG-SC groups is observed.

### 2.4. OEMSCs Increase the Number of Axons in the Delayed Animal Model

When the damaged nerve is immediately vein-bridged, stem cell grafting (IVG-SC) triggers various processes and leads to a significant increase in the number of axons in the medial (*p* < 0.001)) and distal (*p* < 0.001) parts of the nerve, when compared to the Control group (Figure 4A). A similar result is observed when comparing the ungrafted IVG group to the Control group (medial, *p* < 0.05; distal, *p* < 0.01). In addition, statistical analysis indicates that the number of axons is significantly higher in the medial (*p* < 0.01) and distal (*p* < 0.01) areas for stem cell-transplanted animals (IVG-SC) when compared to ungrafted rats (IVG). When the insertion of the vein is delayed, stem cells grafting (DVG-SC) increases the number of axons in the medial (*p* < 0.01) and distal (*p* < 0.01) areas, when compared to the Control group. For the DVG animals, compared to the Control group, a significant difference is only reported in the medial (*p* < 0.001) area (Figure 4B). As a result, the number of axons in the DVG-SC is significantly higher (*p* < 0.05) in the distal area when compared to the value of the DVG group. No difference between the IVG-SC and the DVG-SC groups is observed. Conversely, a significant difference (*p* < 0.05) is reported in the distal area when comparing the IVG and the DVG groups. The number of axons/mm^2^ was also calculated. In absence of a significant difference with the results reported above, they are not displayed.

### 2.5. OEMSCs Increase the Total Amount of Myelin but Not the G Ratio

Rats with an inserted vein, filled or not with stem cells, display a significant (IVG, *p* < 0.001; IVG-SC, *p* < 0.01; DVG, *p* < 0.01; DVG-SC, *p* < 0.001) increased myelination, when compared to the Control animals (Figure 4C,D). No difference is observed between the IVG and the IVG-SC groups but a significant difference is noticed when comparing DVG and DVG-SC groups (*p* < 0.05). Furthermore, no difference between the IVG-SC and the DVG-SC groups is observed but a reduced percentage of myelin is reported in the DVG group compared to the IVG group (*p* < 0.05). However, this increased amount of lipids in the operated groups is not associated to an augmented myelin sheath around individual axons since the G ratio, defined as the ratio between axon diameter and myelinated fiber outer diameter remains unchanged (Figure 4E,F). No difference between IVG vs. DVG and IVG-SC vs. DVG-SC groups is observed.

### 2.6. OEMSCs Do Not Migrate Outside the Implanted Tissue

Grafting stem cells can be a contentious issue. For example, it can be hypothesized that these highly proliferative cells invade other body parts and, possibly, trigger tumor formation. As a result, preclinical studies must prove that they do not migrate outside the implanted tissue/organ. For that purpose, we transplanted male stem cells into female hosts. Using the quantitative PCR technique, we assessed the presence of male cells into the repaired peroneal nerve but also in the brain, heart, kidney, liver, and lung. As shown on the Appendix A, the expression of the gene *sry* (sex determining region of Y chromosome) is solely found in the grafted nerve.

## 3. Discussion

The goal of the present study was to assess the muscle properties, the metabosensitive and locomotor recovery of animals with a transected peroneal nerve, benefiting from an acute or delayed implantation of a vein, filled or not with nasal olfactory stem cells. We show here that, three months after the trauma, the dramatic consequences of a 3 mm nerve loss are partially overcome with the acute insertion of a venous bridge. Interestingly, when the repair strategy is delayed for 2 weeks, a similar outcome is achieved. In addition, cell grafting induces an impressive axon sprouting, in the distal and medial areas, in both the acute and delayed modes of repair.

### 3.1. The Vein, a Suitable Biomaterial for Nerve Regeneration

The observed locomotor recovery, three months after the initial trauma, is one of the most remarkable findings of the current study. In the acute condition, the inserted vein offers a permissive environment that induces a full locomotor recovery at W12, a result in accordance with previous studies [7,8,9,11,30,31,32]. Conversely, a delayed vein insertion leads to an unsatisfactory recovery, as formerly observed in three studies [33,34,35]. It must however be mentioned that, in clinical practice, the vein bridging of neglected wounds, even two years after the initial insult, triggers a surprisingly good healing [7,36,37]. The most appropriate functional outcome to be measured, according to the animal species, the type of injured nerve and the nature of the experiment (pre-clinical versus clinical), can be found in a recent review [38].

### 3.2. Muscle Mass Is Not Correlated to Function

Unexpectedly, the improved locomotion is not correlated to a preserved or rescued muscle mass. As pointed out hitherto, the muscle weight cannot in itself be considered as an indicator of muscle recovery since the latter is also influenced by other factors, such as the amount of intramuscular connective tissue [11]. Therefore, in future experiments, analysis of muscle weight should be supplemented with histological and morphometric analyses of muscle fibers. In addition, as previously reported, the longer the denervation period, the less effective the re-innervation is [39,40]. It is also important to note that, in neglected wounds, the restoration of muscle function depends on the number of axons reaching their terminal targets but also on their ability to form new neuromuscular junctions.

The evoked muscle twitch provides information on the contractile properties, as well as the quality of the regenerated nerve efferents. It is well established that, after a section of the peroneal nerve, the tibialis anterior muscle, considered as a so-called “fast” muscle, becomes partly atrophied and turns into a so-called “slow” muscle [41]. Such a process is observed when a stem cell-free vein is inserted. In other words, the vein is a permissive environment but not sufficient in itself to allow the muscle to partly regain its initial phenotype. Fortunately, the implantation of stem cells helps to improve the locomotor recovery.

Finally, we can hypothesize that, at the time of analysis (3 months post-surgery), all nerve fibers that regenerated and reached the target muscle are not yet mature enough to lead to a recovery of muscle mass or phenotype. This may explain the lack of correlation between function and muscle properties. To speed the process of regeneration, it could be envisioned to use electrical stimulation, a technique that has been successfully used in animal models and humans, grafted with autologous nerve segments (for a review [42]).

### 3.3. Olfactory Stem Cells, a Positive Add-On for Neglected Wounds

In both arms of the study, the grafting of stem cells proved to be beneficial for the injured animals. When implanted, OEMSCs (i) quicken locomotor recovery, (ii) improve muscle mechanical properties and (iii) increase the number of growing axons.

In regard to the repair of the nervous systems, the current findings are in line with previous reports. For example, it was observed that, 11 weeks after a partial cervical hemisection, human lamina propria-derived neural progenitors improve spontaneous vertical exploration and horizontal rope walking [25]. Later on, the same team, using the same animal model, demonstrated that human olfactory stem cells enhance the functional recovery of the forelimb [26].

In regard to peripheral nerve repair, it has been shown that nasal olfactory stem cells, transplanted in a biphasic collagen and laminin guidance conduit, across a 1 cm sciatic nerve gap, improved axogenesis and reduced nociception [29]. More recently, it was shown that, three months after their transplantation in a rat model of facial nerve injury, OEMSCs induce functional recovery. They significantly increase (i) maximal amplitude of vibrissae protraction and retraction cycles and (ii) angular velocity during protraction of vibrissae [43]. Of note, an olfactory bulb-associated cell type, namely the olfactory ensheathing cells, promotes facial regeneration and functional recovery when implanted in a rat transected facial nerve [44].

Such improved behaviors are usually associated with an increased axogenesis or a limited Wallerian degeneration. In the present study, an obvious stem cell-related axon elongation/sprouting is at play. Among the putative underlying molecular mechanisms, we can cite the cytokines secreted by OEMSCs. Two independent studies, based on mass spectrometry, listed the content of OEMSC secretome [23,45]. The first one identified 274 proteins, of which 45 are known to be involved in cell growth, migration and differentiation [45]; the second listed 629 proteins, including 43 and 32 molecules associated to axonal guidance signaling and actin cytoskeleton signaling, respectively [23]. Top candidates include Netrin 4 (NTN4), Platelet Derived Growth Factor C (PDGFC), Transforming Growth Factor Beta1 (TGFB1), and Transforming Growth Factor Beta 2 (TGFB2).

In addition to trophic molecules, OEMSCs also secrete proteins related to the immune system. The two above-mentioned studies [23,45] reveal that inflammation associated molecules are observed in the secretome, notably several interleukins and members of the HLA complex. Overall, as described in two articles [21,46], OEMSCs exert immune-modulatory functions. A strong anti-apoptotic effect toward not-activated immune effector cells is displayed by OEMSCs, in comparison to other adult stem cells originating from various organs [21]. More specifically, OEMSCs boost regulatory T (Treg) cell differentiation and impede T cell proliferation [46]. In sum, the enhanced axogenesis observed in the current study, may also be the result of a stem cell-associated reduced inflammation.

### 3.4. Clinical Applications

As most pre-clinical studies focusing on peripheral nerve regeneration, we decided to assess the animals during three months after the injury. It would have been of interest to extend this post-trauma period in order to possibly observe a more comprehensive recovery. For example, we noticed a high number of axons in the medial and distal parts of the nerve. This finding is in line with an early observation indicating that, in a rat model of nerve repair, the number of distal axons may increase significantly in the first few months and reach a twofold value after three months. However, axons failing to reach their appropriate terminal targets get atrophied over time and, two years after, near-normal values are observed [47]. Similarly, it could be envisaged that, in the mid- or long-term, the tibialis anterior muscle recovers its initial mass and characteristics due to nerve fiber maturation.

The current study was mainly designed to respond to a clinical issue, the repair of neglected wounds. We opted for a two-week delay between the loss of nerve substance and the insertion of vein and cells. Our protocol was based on the minimum time necessary to produce a relatively large quantity of stem cells in GMP conditions and, therefore, allow autologous transplantations. Nevertheless, some nerve injuries can be overlooked for several years [48,49] and it may be wise to assess our therapeutic strategy in long-lasting nerve injuries.

Like in many cell therapy experiments, we implanted stem cells in order to improve functional recovery. For many years, it was thought that the main role of these undifferentiated cells was to turn into differentiated cells, the ones required by the microenvironment they were implanted into. However, as mentioned above, it is now well established that they also provide trophic and immuno-modulatory molecules [21,23,45,46,50]. To test this hypothesis, our study could have included additional experimental groups in which the vein would have been filled with extracellular vesicles released from stem cells, as demonstrated by a team using exosomes from olfactory ensheathing cells [51]. Interestingly, extracellular vesicles can be used autologously or heterologously (without the use of immune-suppresants) and recurrently injected, locally or intravenously when the repair remains unsatisfactory.

## 4. Materials and Methods

### 4.1. Animals

Six-week-old female Fischer rats (*n* = 67), weighing 250–300 g (Janvier Labs^®^, Le Genest-Saint-Isle, France), were housed in smooth-bottomed plastic cages at 22 °C under 12 h light/dark cycle. Food (Purina^®^, rat chow, St. Louis, MI, USA) and water were available *ad libitum*. All animals were weighed before each experiment. Anesthesia and surgical procedures were performed according to the French law (Decrees and orders N°2013-118 of 1 February 2013, JORF n°0032) on animal care guidelines and after approval by animal Care Committees of Aix-Marseille Université (AMU) and Centre National de la Recherche Scientifique (CNRS). All individuals conducting the research were listed in the authorized personnel section of the animal research protocol (License n°B13.013.06). Furthermore, experiments were performed following the recommendations provided in the Guide for Care and Use of Laboratory Animals (U.S. Department of Health and Human Services, National Institutes of Health) and in accordance with the directives 86/609/EEC and 010/63/EU of the European Parliament and of the Council of 24 November 1986 and of 22 September 2010, respectively, and the ARRIVE (Animal Research: Reporting of In Vivo Experiments) guidelines. Any animal presenting sign of suffering such as screech, prostration, hyperactivity, anorexia, and paw-eating behavior were sacrificed.

### 4.2. Experimental Groups and Surgeries

Rats were randomized into five groups: (1) Control group (*n* = 8) in which no surgery was performed, (2) immediate vein graft (IVG) group (*n* = 14) in which a segment of 3 mm peroneal nerve was removed and a vein conduit of 1 cm was immediately grafted between the two nerve stumps and filled with 10 μL of thrombin/fibrinogen, (3) immediate vein graft and stem cells (IVG-SC) group (*n* = 13) in which a segment of 3 mm peroneal nerve was removed, a vein conduit of 1 cm was immediately grafted between the two nerve stumps and filled two weeks later with one million stem cells resuspended in 10 μL of thrombin/fibrinogen, (4) delayed vein graft (DVG) group (*n* = 10) in which a segment of 3 mm peroneal nerve was removed and a vein conduit of 1 cm was grafted, two weeks later, between the two nerve stumps, and filled with 10 μL of thrombin/fibrinogen, (5) delayed vein graft and stem cells (DVG-SC) group (*n* = 13) in which a segment of 3 mm peroneal nerve was removed and a vein conduit of 1 cm, filled with one million stem cells, resuspended in 10 μL of thrombin/fibrinogen, was grafted, two weeks later, between the nerve stumps. In addition, 9 DVG-SC rats were included in order to assess the putative presence of stem cells in other organs.

### 4.3. Surgery Procedure

Animals were deeply anesthetized (isoflurane 3%, Isoflurin^®^, Axience Santé Animale SAS, Pantin, France) and surgical procedures were performed aseptically under binoculars. For the IVG, IVG-SC, DVG, and DVG-SC groups, the peroneal nerve from the left limb was dissected free from the surrounding tissues and cut on a 3 mm length, 5 mm away from the sciatic nerve trifurcation (i.e., starting point of tibial nerve, common peroneal nerve, and caudal sural cutaneous nerve). A1 cm long femoral vein was removed from the contralateral side of the nerve injury, washed in saline (NaCl 0.9%) and grafted immediately. The two nerve stumps were inserted into the vein, leaving a gap of 3 mm between the proximal and distal nerve stumps. The graft was fixed using three or four 9-0 monofilament non-absorbable sutures (Ethilon^®^ 9-0, Ethicon Inc. Jonhson& Johnson, Sommerville, NJ., USA) for each stump as well as biological thrombin/fibrinogen glue (Tisseel, Baxter). For the IVG-SC and DVG-SC groups, one million of OEMSCs were resuspended in 5 μL of thrombin, injected in the vein (IVG-SC andDVG-SC groups), using a Hamilton syringe (Hamilton Company, Bonaduz, Switzerland), filled with 5 μL of fibrinogen. Stem cells were implanted two weeks after the vein insertion for the IVG-SC group and at the time of vein insertion for the DVG-SC group. In other words, in both cases, cells were grafted two weeks after the initial surgery. Muscles and skin were immediately stitched using non-absorbable monofilament sutures (Ethilon^®^ 3-0, Ethicon Inc. Jonhson & Johnson, Sommerville, NJ, USA).

Clinical signs of pain or unpleasant sensations (self-mutilation, prostration, hyperactivity, anorexia, and weight loss) were examined daily. All animals were treated with an injection of buprenorphine (0.5 mg/kg subcutaneously) post-surgery. No other administration of this antalgic was performed, indicating that no sign of pain was observed.

### 4.4. Cell Culture

OEMSCs were cultivated according to a previously described protocol [24,52], with slight modifications. Briefly, olfactory mucosae, lying on the nasal septum, were collected from 8 adult male rats and transferred in a DMEM/HAM F12-filled Petri dish. After a triple washing for eliminating the mucus, the biopsies were incubated in a Petri dish filled with 1 mL of dispase II solution (2.4 IU/mL), for 1 h at 37 °C. Next, the olfactory epithelium was removed from the underlying lamina propria using a micro spatula. Once purified, the lamina propria was cut into small pieces with two 25 gauge needles and transferred into a 15 mL tube filled with 1 mL of collagenase NB5 (1U/mL, Nordmark Biochemicals). After a 10 min incubation at 37 °C, the tissue was mechanically dissociated and the enzymatic activity was stopped by adding 9 mL of Ca-free and Mg-free PBS. After centrifugation at 200× *g* for 5 min, the cell pellet was resuspended in DMEM/HAM F12 culture medium, supplemented with penicillin/streptomycin and 10% platelet lysate (Macopharma^®^, Tourcoing, France) and plated on plastic culture dishes. The culture medium was renewed every 2 to 3 days.

Before being implanted, cultivated OEMSCs were characterized using fluorescent antibodies designed for flow cytometry assays. Cells were trypsinized, centrifuged, and resuspended in cold PBS. One hundred thousand cells were incubated 30 min at 4 °C in cold PBS (200 μL) with phycoerythrin (PE)-labeled monoclonal antibodies, added at saturating concentration. The antibodies used were: PE Mouse Anti-Rat CD45 (#554878, BD Biosciences), Mouse Anti-Rat CD73 (#551123, BD Biosciences), PE Mouse Anti-Rat CD90 (#551401, BD Biosciences), PE Mouse Anti-Nestin (#561230, BD Biosciences). Cells were then washed twice by centrifugation, resuspended with cold PBS, and then processed immediately for flow cytometric analysis.

### 4.5. Functional Assessment of Hind Limb Recovery

In order to measure functional alteration, footprints were recorded every two weeks (W4, W6, W8, W10, and W12) on paper track and were analyzed as previously described [53]. Briefly, the peroneal functional index (PFI)was calculated by the method of Bain et al. [54] with the following formula: PFI = 174.9 × [(ePL-nPL)/nPL] + 80.3 × [(eTS-nTS)/nTS] − 13.4. The parameters measured for both normal (n) and operated (e) feet were footprint length (PL, or longitudinal distance between the tip of the longest toe and the heel) and total toe spreading (TS, or cross-sectional distance between the first and fifth toes). The recovery rate of the PFI was defined on a score ranging from −100 to −13.4, where −13.4 represents normal function and −100 total failure. Footprints were obtained from the second to the 12th week post-surgery. Before surgery, all animals were conditioned to walk, during two weeks, in the corridor, two times per day and five days per week (PRE-values).

### 4.6. Electrophysiological Recordings

Twelve weeks after surgery, the rats were anesthetized by intraperitoneal injection of urethane solution (i.p. 1.2 g/kg, Sigma Aldrich-Merck Merck KGaA, Darmstadt, Germany). A tracheotomy was performed and a cannula equipped with a thermocouple measuring changes in airflow temperature was inserted into the trachea. The left peroneal nerve was dissected and removed from the surrounding tissues over a length of 3–4 cm. A catheter was inserted into the right femoral artery and pushed to the bifurcation of the descending abdominal aorta to transport chemicals (potassium chloride, KCl (20 mM in 0.5 mL of saline) and lactic acid, LA (1 mM in 0.1 mL of saline) to the contralateral muscle. This catheter was used to allow the blood to flow freely into the muscles of the lower left limb.

#### 4.6.1. Twitch Measurement

Two stimulation electrodes (inter-electrode distance: 1 mm) were placed on the peroneal nerve surface. The contractile response of the tibialis anterior to nerve stimulation (twitch which is a reflection of the tension generated in the muscle) was induced with a neurostimulator (rectangular single shocks, duration: 0.1 ms, frequency: 0.5 Hz delivered by a S88K stimulator, Grass Instruments/Astro Med Inc., West Warwick, RI, USA) and measured with anisometric strain gauge (micromanometer 7001, Ugo Basile, Ugo Basile SRL, Gemonio, Italy). Several parameters were recorded: amplitude (A), maximum contraction rate (MCR), maximum relaxation rate (MRR), defined as the slope of a tangent drawn towards the steepest part of the contraction or relaxation curve. MCR and MRR were normalized to the amplitude of twitch (MCR/A and MRR/A, msec^−1^). Twitch was recorded with the Biopac MP150 system (sampled at 2000 Hz, filtered with low pass at 150 Hz) and analyzed using the AcqKnowledge 3.7.3 software (Biopac Systems Inc., Goleta, CA, USA).

#### 4.6.2. Ventilatory Response

Changes in ventilation were recorded after tibialis anterior stimulation as demonstrated by a previous study [55], showing that repetitive muscle stimulation induces fatigue, which activates the muscle metabosensitive afferent fibers projecting to the bulbar respiratory center and subsequently, increases ventilation. To elicit electrically-induced muscle fatigue (EIF), the S88K stimulator was used to induce rhythmic muscle contractions that delivered pulse trains to the muscle surface electrode (pulse duration: 0.1 ms; frequency: 10 Hz, i.e., 5 shocks in each 500 ms train; duty cycle: 500/1500 ms). The voltage was supramaximal, i.e., 20% higher than that used to elicit a maximal twitch. Fatigue was assessed from the decay of force throughout the 3-min EIF period. The muscle was directly stimulated because it has been shown that muscle low frequency stimulation is a strong activator of metabosensitive afferent fibers [56]. Ventilatory activity was recorded through the tracheal cannula 2 min before EIF (rest condition) and 3 min after and expressed in cycles/min.

#### 4.6.3. Afferent Activity

In order to record the afferent activity from the tibialis anterior muscle, nerve was positioned on a monopolar tungsten electrode and immersed in paraffin oil. Nerve activity was recorded in reference to a nearby ground electrode implanted in a closed muscle, amplified (50 to 100 K) and filtered (30 Hz to 10 kHz) by a differential amplifier (P2MP^®^ SARL, Marseille, France). The afferent discharge was recorded (Biopac MP150) and fed into pulse window discriminators (P2MP^®^ SARL, Marseille, France), analyzing simultaneously afferent activity. The output of these discriminators provided noise-free tracings (discriminated units) that were counted by a data analysis system (Biopac AcqKnowledge software) at 1 s intervals (in Hz) and then displayed on a computer. The discriminated units were counted and recorded on separate tracings.

As previously described [57], we recorded the response of muscle afferents after (1) 3-min EIF period, (2) intra-arterial bolus injection of KCl (20 mM in 0.5 mL of saline) or LA (1 mM in 0.1 mL of saline) solutions. The discharge rate of nerve afferents was averaged for a 30s period before and after EIF or metabolite injection (baseline discharge). The increase in average afferent discharge rate after EIF or metabolite injection was expressed as a percentage of the average of the afferent discharge rates before activation. A 20 min recovery period was allowed after EIF and between LA and KCL injections.

### 4.7. Muscular Atrophy Measurement

After the electrophysiological recording, rats were sacrificed by an intra-arterial overdose (1 mL) of sodium pentobarbital solution (Pentobarbital Sodique, Sanofi Santé Animale, 60 mg/mL). Left tibialis anterior muscle was harvested and immediately weighed and the muscle weight/body weight ratio was calculated.

### 4.8. RNA Purification and Quantitative Real-Time PCR

Twelve weeks after the initial surgery, brain, heart, liver, lungs, kidneys and peroneal nerve were collected from female host rats that were either grafted or not with male olfactory stem cells (*n* = 9 per group for grafted animals and *n* = 3 for the control group). Total RNA was isolated using RNeasy Kit (Qiagen, Courtaboeuf, France), according to the manufacturer’s instructions.

Total RNA was reverse transcribed with the High-Capacity RNA-to-cDNA Kit (Applied Biosystems, Thermo Fisher Scientific, Villebon sur Yvette, France) following manufacturer’s instructions and using a Veriti™ 96-Well thermal cycler (Applied Biosystems). Real time quantitative PCRs were carried out to detect the expression of the molecular target of interest using a combination of specific primers to detect *sry* gene. *TBP* (TATA box binding protein) was chosen as a housekeeping gene and its expression levels served as reference. The real time PCR reactions were carried out with 12.5 ng of cDNA in a mix solution containing 1X iTaq™ Universal SYBR^®^Green Supermix (Bio-Rad, Hercules, CA, USA) and 300 nM of forward and reverse primers. Oligonucleotides were designed from the nucleotide sequences described in GenBank: *sry* forward primer: 5′-GCGCAAGTTGGCTCAACAGAAT-3′; sry reverse primer: 5′-TCGGCTTCTGTAAGGCTTTTCCAC-3′; *TBP* forward primer: 5′-TGGGCTTCCCAGCTAAGTTCTTAGAC-3′; *TBP* reverse primer: 5′-TTCCAGCCTTATGGGGAACTTCAC-3′.

Samples were amplified in triplicates. Mean Ct for each sample was calculated, and relative expression levels were determined according to the ∆∆Ct method: ∆Ct values represent the normalized levels of each target gene compared to the *TBP* control, and ∆∆Ct values were calculated by subtracting the mean ∆Ct of the control population to the sample ∆Ct. Relative expression levels were calculated using the 2^−∆∆*C*t^ equation. Negative fold change values were obtained by the equation 1/(mean 2^−∆∆*C*t^) of each group.

### 4.9. Histology, Immunocytochemistry and Microscopy

Peroneal nerves were collected, rapidly washed in phosphate buffer (PBS 1X, Gibco^®^, Thermo Fisher Scientific, Villebon sur Yvette, France) and fixed with 4% paraformaldehyde (Antigenfix solution, Diapath S.P.A, Martinengo, Italy) for 24 h. Then, nerves were sectioned in three parts (proximal end, medial part, and distal end) and stored at −20 °C until used. For each group of animals, samples were included in paraffin. After embedding, sections of 5 µm thick were performed using a microtome (RM2155, Leica Biosystems^®^, Wetzlar, Germany) and collected on coated slides (TOMO^®^, VWR International, Matsunami Glass IND Ltd., Osaka, Japan). Sections were dewaxed in three changes of Histo-Choice^®^ (Sigma Aldrich-Merck) for 10 min each, rehydrated in 100% ethanol, 95% ethanol, 70% ethanol, 50% ethanol (twice in each 10 min), and rinsed in two changes of distilled water for 5 min each.

For axon quantification, nerve sections (distal, medial, proximal) were immunostained with a mouse monoclonal antibody raised against the light chain of neurofilament protein (NF-L 70 KDa, Sigma Aldrich-Merck, dilution: 1:500) using a robot (Benchmark^®^ XT, Roche Diagnostics, F. Hoffmann-La Roche Ltd., Basel, Switzerland). After washing, an appropriate biotinylated-conjugated secondary antibody was applied to the sections. The final staining step was performed using diaminobenzidine (iView DABDetection Kit, Ventana Medical Systems Inc., Oro Valley, AZ, USA).

For myelin quantification, only medial nervesections were used. An antigen retrieval procedure was carried out. Slides were immersed in sodium citrate buffer (10 mM sodium citrate, 0.05% Tween-20, pH 6.0) for 20 min at 95 °C, then progressively cooled down for 20 min. After a washing step in PBS, sections were blocked for 1 h at room temperature in a blocking buffer containing 0, 3% Triton X-100, 3% bovine serum albumin (BSA) and 10% PBS 10X. The medial peroneal nerves sections were immunostained with a rabbit monoclonal antibody raised against the myelin protein zero (MPZ ab183868, dilution: 1:100, Abcam Plc, Cambridge, UK). After washing, an appropriate biotinylated-conjugated secondary antibody was applied to the sections (goat anti-rabbit IgG biotin, Abcam Plc, dilution 1:500). The final staining step was performed using 1% diaminobenzidine (DAB, Sigma Aldrich-Merck), 1% NiCl and 1% H_2_O_2_ in PBS.

In addition, using alternate sections, myelin was also stained using the conventional Luxol method. Nerve sections were incubated overnight at 56 °C in 0.1% Luxol (Clinisciences, Nanterre, France) solution (in 95% ethanol, 5% glacial acid acetic). After two washings in 95% ethanol and distilled water, sections were differentiated 30 s first, in a 0.05% Lithium carbonate solution followed by another differentiation step in 70% ethanol for 30 s. After washing in distilled water, sections were incubated in 95% ethanol for 5 min, in 100% ethanol twice for 5 min, and finally in Histo-Choice^®^ (Sigma Aldrich-Merck) twice for 5 min.

Stained sections were examined using an Apotome microscope (Apotome v2, AxioObserver Z1, Zeiss) which was associated with high-resolution camera (Caméra CMOS Orca flash 4.0 v2, Zeiss, Oberkochen, Germany). The slides were digitized and analyzed with ImageJ (NIH) software. Axon numbers, axon areas and myelin content were measured in each group of animals. To assess G-ratio (i.e., the ratio between the diameter of the axon and the outer diameter of the myelinated fiber), slides were coded, 5 regions of interest in each section were randomly chosen and data analysis (100 fibers per group) was performed blindly at ×100 magnification.

### 4.10. Statistics

Results obtained from behavioral locomotor tests (PFI), electrophysiological recordings (EIF, KCl and lactate injection), muscle properties (weight/body weight ratio, contraction properties), and histological data were compared between all experimental groups. Data processing was performed using a software program (SigmaStat^®^, San Jose, CA, USA). Normal data distribution was verified (Shapiro–Wilk test) and a two-way ANOVA (group factor × time factor) for repeated measures was used to compare groups with each other and over time for behavioral scores. One-way ANOVA (group factor) were used to determine the electrophysiological responses, and for the histological analysis. Then, statistics were completed with a multiple-comparison post-hoc test (Student- Newman–Keuls method).Data were expressed as mean±SEM. Difference was considered significant when *p* < 0.05.

## 5. Conclusions

The current study shows that the insertion of a nerve conduit, filled with nasal olfactory stem cells, greatly improves the functional recovery of rats with a peroneal nerve injury, especially when the repair strategy is delayed. These promising results open the way for a phase I/IIa clinical trial based on the autologous grafting of OEMSCs in patients with a neglected peripheral nerve injury.

## Figures and Tables

**Figure 1 ijms-21-02670-f001:**
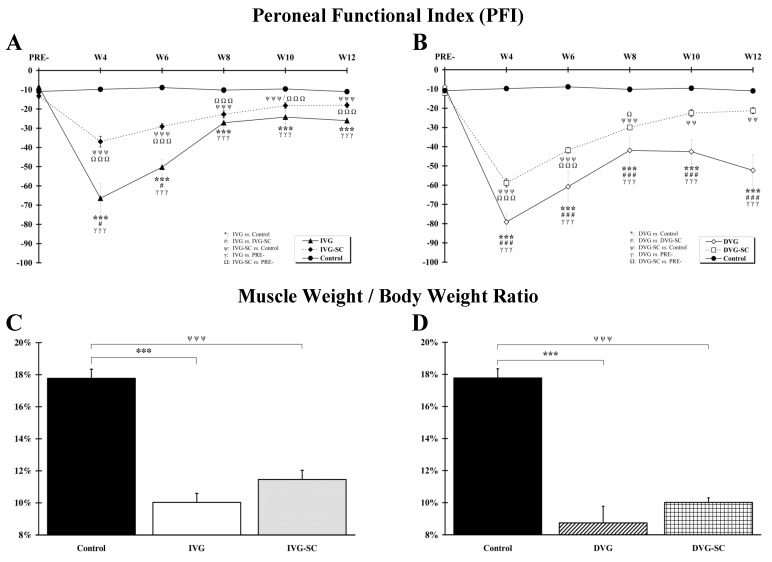
Peroneal Functional Index (PFI) and relative weight of the tibialis anterior muscle. (**A**,**B**). The PFI was measured every two weeks, from W4 to W12 post-surgery. When the vein was immediately grafted, (**A**) an improvement was observed at W4 and W6 for the stem cell-grafted animals compared to ungrafted animals (immediate vein graft and stem cells (IVG-SC) vs. immediate vein graft (IVG)). When the vein was grafted two weeks after the injury (**B**) and filled with stem cells, an improvement was observed from W4 until W12. (**C**,**D**). The relative weight of the tibialis anterior was assessed twelve weeks post-surgery. The muscle weight/body weight ratio in the IVG and IVG-SC groups was significantly reduced when compared to the Control group (**C**). 1 symbol: *p* < 0.05; 2 symbols: *p* < 0.01; 3 symbols: *p* < 0.001.

**Figure 2 ijms-21-02670-f002:**
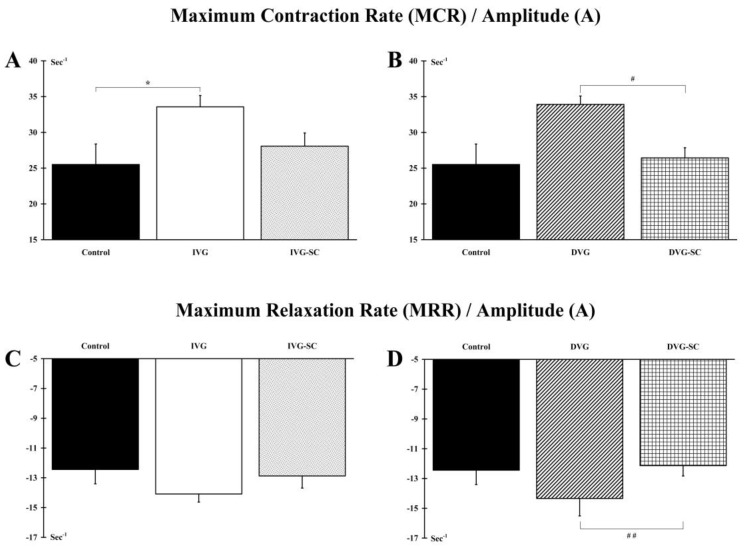
Muscle mechanical properties. (**A**,**B**) IVG and DVG groups displayed a significant increase in the MCR/A ratio compared to Control and DVG-SC groups, respectively. Conversely, stem cells allowed the IVG-SC and DVG-SC groups to maintain a phenotype close to the Control group. (**C**,**D**) Concerning the MRR/A ratio, animals from the IVG, DVG, IVG-SC and DVG-SC groups displayed a ratio similar to the Control group while DVG group exhibited a reduced ratio when compared to the DVG-SC group. *: IVG group vs. Control group; #: DVG group vs. DVG-SC group; 1 symbol: *p* < 0.05, 2 symbols: *p* < 0.01.

**Figure 3 ijms-21-02670-f003:**
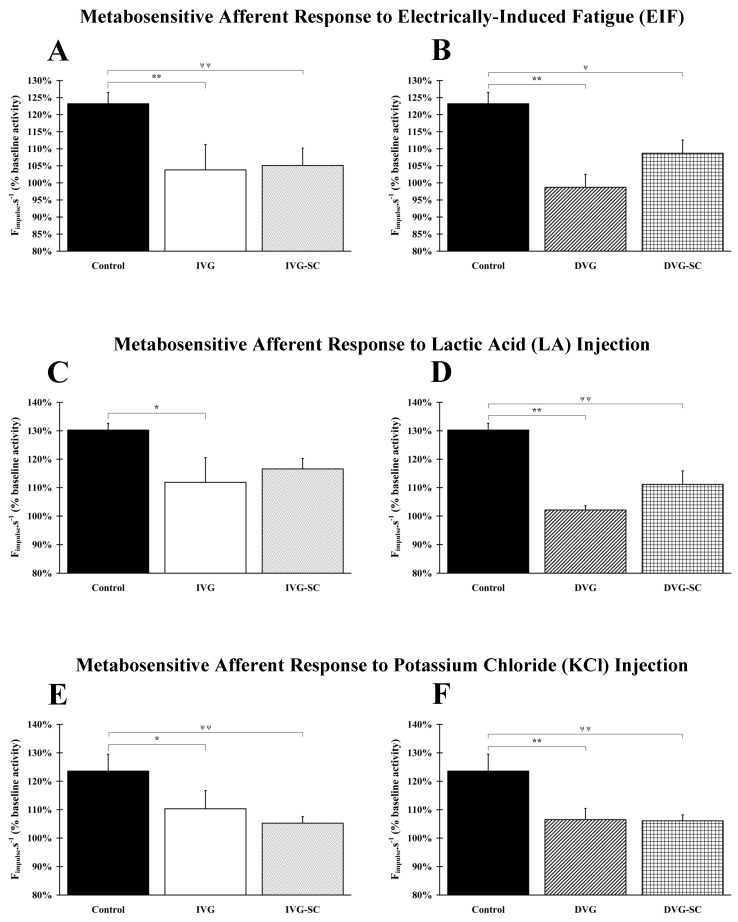
Nerve response to afferent stimulation. (**A**,**B**) Compared to Control group, responses associated to electrically-induced fatigue were significantly lower in all vein-bridged animals, even when stem cells were added in the vein chamber. (**C**,**D**) A significant decrease of afferent response was observed in the IVG, DVG, and DVG-SC groups, after injection of lactic acid. No significant difference was observed between the IVG-SC group and the Control group. (**E**,**F**) A significant decrease of afferent response was observed in all groups after injection of potassium chloride. Grafting of stem cells does not improve the recovery. *: IVG or DVG group vs. Control group; ψ: IVG-SC or DVG-SC vs. Control group. 1 symbol: *p* < 0.05; 2 symbols: *p* < 0.01.

**Figure 4 ijms-21-02670-f004:**
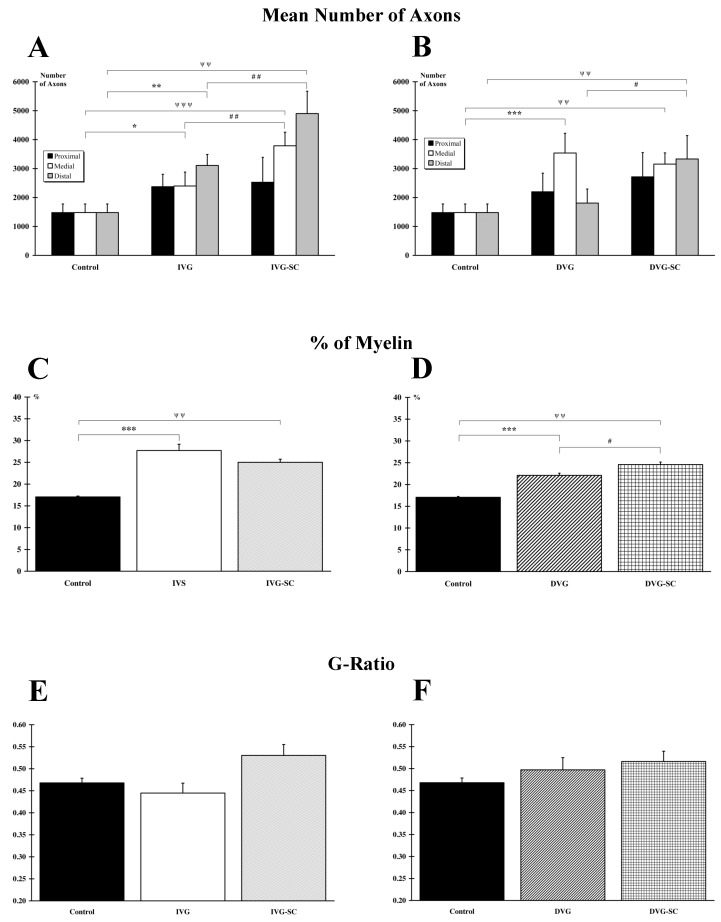
Histological analysis of the peroneal nerve. (**A**,**B**) Quantitative analysis of axon number in each group and at the proximal, medial (vein bridge) and distal parts. In the IVG and IVG-SC groups the number of axons increased in the medial and distal parts, compared to the Control group (**A**). In the DVG and DVG-SC groups, the number of axons increased in the medial part. Only the DVG-SC group presented an increased number in the distal part (**B**). (**C**,**D**) Analysis of the percentage of myelin indicates that all repaired groups exhibited a higher percentage of myelin compared to Control group. However, the DVG-SC group displayed a higher number than the DVG group (**D**). (**E**,**F**) No difference between groups is observed when analyzing the G-ratio.*: IVG or DVG group vs. Control group; #: IVG or DVG group vs. IVG-SC or DVG-SC group, respectively; ψ: IVG-SC or DVG-SC vs. Control group. 1 symbol: *p* < 0.05; 2 symbols: *p* < 0.01; 3 symbols: *p* < 0.001.

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
