# Peer review of "Immediate or Delayed Transplantation of a Vein Conduit Filled with Nasal Olfactory Stem Cells Improves Locomotion and Axogenesis in Rats after a Peroneal Nerve Loss of Substance"

_ijms, 2020, doi:10.3390/ijms21082670_

Round 1

Reviewer 1 Report

Bonnet M., et al. presented the therapeutic potential of vein guides, transplanted immediately or two weeks after a peroneal nerve injury and filled with olfactory ecto-mesenchymal stem cells (OEMSC). The authors used rats for 3 mm peroneal nerve injury model. The authors insisted that the transplantation with OEMSC improved locomotion and axogenesis in rat models with peroneal nerve loss.

The study designs is appropriate and the authors used valid experimental methods for analysis. The interpretations of the results are supported by the data. However, the authors should make more informative to the manuscript.

  1. In Materials and methods (3 page), the number of stem cells inserted with vein grafts might be better described in 2.2. experimental groups and surgeries, the authors mentioned it 2.3 surgical procedure, though.
  2. I agree the authors' opinion to analysis of muscle weight, the muscle weight itself could not be an indicator of muscle recovery. However the authors did not analyse the muscle fibers histologically or morphologically in this study.  
  3. In discussion, the authors should provided some important proteins and molecules associated with axogenesis, although the authors gave the references.        
  4. The authors mentioned a phase I/IIa clinical trial in abstract and conclusion sections. However, there is no information about the clinical trial. Please provide the progress of this clinical trial in detail.
  5. Please revise or adjust  the size of the letters in Figures. Especially Figures 3 and 4, the letters along with x-axis  are so small.    

Author Response

The study designs is appropriate and the authors used valid experimental methods for analysis. The interpretations of the results are supported by the data. However, the authors should make more informative to the manuscript.

  1. In Materials and methods (3 page), the number of stem cells inserted with vein grafts might be better described in 2.2. experimental groups and surgeries, the authors mentioned it 2.3 surgical procedure, though.

Response: We are sorry for not having been more precise on this point. We inserted one million cells in each vein. The text now reads:

Rats were randomized into five groups: 1) Control group (n=8) in which no surgery was performed, 2) Immediate Vein Graft (IVG) group (n=14) in which a segment of 3 mm peroneal nerve was removed and a vein conduit of 1 cm was immediately grafted between the two nerve stumps, 3) Immediate Vein Graft and Stem Cells (IVG-SC) group (n=13) in which a segment of 3 mm peroneal nerve was removed, a vein conduit of 1 cm was immediately grafted between the two nerve stumps and filled two weeks later with one million stem cells, 4) Delayed Vein Graft (DVG) group (n=10) in which a segment of 3 mm peroneal nerve was removed and a vein conduit of 1 cm was grafted, two weeks later, between the two nerve stumps, 5) Delayed Vein Graft and Stem Cells (DVG-SC) group (n=22) in which a segment of 3 mm peroneal nerve was removed and a vein conduit of 1 cm, filled with one million stem cells, was grafted, two weeks later, between the nerve stumps.

  1. I agree the authors' opinion to analysis of muscle weight, the muscle weight itself could not be an indicator of muscle recovery. However, the authors did not analyse the muscle fibers histologically or morphologically in this study. 

Response: We agree with the reviewer that the muscle weight itself could not be the only indicator of muscle recovery but as previously described (Pertici et al., 2014a, b; Alluin et al., 2009; Alluin et al., 2006; Marqueste et al., 2004), muscle weight of the lesioned side normalized to the weight of the animal can be an indicator of the muscle atrophy which is related to muscle denervation. When a peripheral nerve is lesioned, the muscle that it innervates gets atrophied. After axonal regeneration, a muscle weight recovery is observed. Evans et al. (1995) proposed that muscle weight measurement is a parameter for functional recovery.

  1. In discussion, the authors should provided some important proteins and molecules associated with axogenesis, although the authors gave the references. 

Response: As suggested by the reviewer, we inserted four examples of proteins associated to axogenesis. The text now reads:

Such improved behaviours are usually associated with an increased axogenesis or a limited Wallerian degeneration. In the present study, an obvious stem cell-related axon elongation/sprouting is at play. Among the putative underlying molecular mechanisms, we can cite the cytokines secreted by OEMSCs. Two independent studies, based on mass spectrometry, listed the content of OEMSC secretome [20,44]. The first one identified 274 proteins, of which 45 are known to be involved in cell growth, migration and differentiation [44]; the second listed 629 proteins, including 43 and 32 molecules associated to axonal guidance signalling and actin cytoskeleton signalling, respectively [20]. Top candidates include Netrin 4 (NTN4), Platelet Derived Growth Factor C (PDGFC), Transforming Growth Factor Beta1 (TGFB1) and Transforming Growth Factor Beta 2 (TGFB2).

-

  1. The authors mentioned a phase I/IIa clinical trial in abstract and conclusion sections. However, there is no information about the clinical trial. Please provide the progress of this clinical trial in detail.

Response: The reviewer is right. We mention a phaseI/IIa clinical trial but only as a perspective, as indicated in the abstract and conclusion. We are currently devising the protocol for this clinical trial but we haven’t got yet the approval of the ethical committee.

  1. Please revise or adjust the size of the letters in Figures. Especially Figures 3 and 4, the letters along with x-axis are so small.   

Response: We thank the reviewer for pointing out this problem. We adjusted the size of the letters in order to make them more readable.

Reviewer 2 Report

The topic of the manuscript – transplantation of stem cells for improving regeneration across nerve defects that have been bridged by biografts – is of general interest to the field of peripheral nerve repair.

The manuscript and study design have, however, some limitations that do substantially limit their scientific and pre-clinical value.

Major points

Introduction

  1. Page 2, first paragraph: what determines the “importance” of a nerve gap? Usually this is it length. The authors should be more precise here.
  2. Page 2, second paragraph: the authors refer to prevcious studies that have been applying vein conduits for peripheral nerve repair, but they do not sufficiently report on the type of nerves that are commonly repaired with this kind of autologous graft. To the knowledge of this reviewer this is mainly digital nerves that present with short defect lengths and that are also often repaired with so-called muscle-in-vein grafts. The authors are asked to provide more specific information regarding the proven applicability of vein or muscle-in-vein conduits. Also it should be mentioned that to some extent similar limitations that are evident for autologous nerve grafts are also limiting the use of autologous vein grafts, e.g. in long gap repair, the availability of long venous grafts maybe considerably limited.
  3. Page 2, paragraph 4: The authors should also refer to literature already available for the use of olfactory ensheathing cells in peripheral nerve repair (e.g. Radtke, Kocsis Int J Mol Sci 2012 13(10) 12911-24, doi: 10.3390/ijms131012911, the review is summarizing some previous studies performed by expert groups – but more studies have been published more recently).
  4. Page 2, paragraph 5: The summary of the study performed by the authors already uncovers the two main limitations. First of all, a 3mm nerve gap is very likely to regenerated spontaneously in the rat sciatic nerve or its branches such as the peroneal nerve investigated in this study. Second, a delay of two weeks does not allow for major changes to have occurred in the denervated distal nerve or its target organs – a minimal period of 6 weeks seems more appropriate to this reviewer, the authors are referred to existing literature, such as the work of Tessa Gordon and her colleagues and other contributions (e.g. Sulaiman OA, Gordon et al. Glia 2000, 32, 234-46 or Neurosurgery 2009,65:A 105-14; Stenberg et al. BMC Neuroscience 2017, 18:53).

Materials and Methods

  1. Section 2.2.: It is not sufficiently clear from this section, why the numbers of animals subjected to each experimental group display such a big variation – from 8 up to 22 animals per group. Also it is not described with enough clarity, when the stem cells have been injected, always two weeks after the initial repair and the repair was eventually delayed for two weeks? The later would have resulted in stem cell injection for the DVG-SC group at a time four weeks after the initial surgery?
  2. The study design lacks an appropriate control group that should have been composed of autologous nerve grafting.
  3. Section 2.3.: It is not clear, how the vein graft was treated in the period from initial harvest until implantation in the DVG and DVG-SC groups. Also it is not sufficiently described, how collapsing of the vein graft was prevented in IVG and DVG animals. Was the vehicle 5µl of each thrombin and fibrinogen also applied to these grafts? How was collapsing then prevented for the DVG-SC during the two weeks before SC injection? A collapse of the vein graft may prevent sufficient regeneration and therefore needs specific consideration.
  4. Section 2.3.: How was pain relieved post surgery?
  5. Section 2.6.: The authors have applied a quite comprehensive electrodiagnostic testing that is, unfortunately, considerably different from the commonly applied protocols (kindly refer to X. Navarro European Journal of Neuroscience, Vol. 43, pp. 271–286, 2016 doi:10.1111/ejn.13033). It is not clear to this reviewer what the benefits of the specific measurements are in comparison to commonly applied techniques. Also it is not clear, why the contralateral healthy nerve or muscle (section 2.7.) could not serve as individual control and why a sham group had to be installed.
  6. Section 2.9.: It is not sufficiently clear what staining protocols were used. Was there a double-staining of myelin applied – detection of P0 protein AND Luxol fast blue staining? Further it is described that DAB was used for viszualising the staining which could have been detected in light microscopy as the luxol staininig, but the authors describe analysis with a TIRF microscope? How exactly were the numbers of axons, axon areas and myelin content determined in the sections? Commonly unbiased stereology and nerve morphometry are performed on tissue that has been post-fixed for myelin detection and embedded in epoxid and subsequently sectioned in ultra-thin nerve cross sections that are then available for light microscopy (e.g. Geuna S., J Comp Neurol 427:333-339).

Results

  1. Page 7, 3rd paragraph: Commonly muscle weight ratio is determined by taking the weight of the muscle from the lesioned and the healthy contralateral leg, why did the authors use a sham control?
  2. Figure 1, B: How do the authors explain the worsening of the PFI performance in the DVG group at w12?
  3. Figure 2 and Figure 3, B/D: How do the authors explain that the DVG-SC group even displayed better recovery than the IVG-SC group – should the degeneration during the delay not have aggravated the conditions prior to repair surgery?
  4. Section 3.5.: How the g-ratio was determined is not sufficiently described.
  5. Representative photomicrographs should be provided for illustrating the histology results.
  6. The authors should comment on the distally increased number of axons they detected in their IVG-SC and DVG-SC groups. Increased sprouting could be valuable at mid-graft level but should not occur pre-dominantly distal to the graft, where axons could form mini-fasicles and neuroma in continuity, the latter could represent a secondary impairment to functional recovery.
  7. Figure 4: Usually axonal counts should be reported as nerve fiber density (number of axons/ square mm) this way also different cross sectional areas are appropriately considered. Where the cross sectional areas not differing between the groups and from proximal to distal along the nerve and the grafts?

Discussion

  1. Section 4.1.: The authors should again more precise in their report of “good healing”. What is the functional outcome to be expected for what type of graft and what kind of nerve to be repaired with the same.
  2. The authors should discuss their findings in the context of the clinical use and pre-clinical evaluation of muscle-in-vein grafts.
  3. Section 4.2.: The conditions to be expected for nerves, Schwann cells, and motor endplates after chronic denervation should be discussed in more detail and also in consideration of the tremendous contribution Tessa Gordon and colleagues have provided to our knowledge.
  4. Page 12, 7th paragraph: There are more reports to be considered with regard to recent publications of similar approaches, e.g. Gu J et al., Neural Regen Res 2019 14 124-31.
  5. Page 13, 1st paragraph: there is more information available on the exosome functions, the authors should consider for their discussion also: Xia B et al. Front Cell Neurosci 13:548.

Minor points

  1. Page 6, 3rd paragraph: does “medial” nerve section stand for what has been described as “middle part” in the first line of the same page?
  2. Legend of Figure 2: “…a significant increase in the …”
  3. Page 10, 2nd paragraph: “For the DVG animals, compared to the Control …”
  4. Page 12, 6th paragraph: “neurites” should be replaced by “axons”

Author Response

The topic of the manuscript – transplantation of stem cells for improving regeneration across nerve defects that have been bridged by biografts – is of general interest to the field of peripheral nerve repair.

The manuscript and study design have, however, some limitations that do substantially limit their scientific and pre-clinical value.

Major points

Introduction

  1. Page 2, first paragraph: what determines the “importance” of a nerve gap? Usually this is it length. The authors should be more precise here.

Response: We agree with the reviewer and modified the sentence which now reads:

Currently, epineural repair is the most appropriate surgical procedure when the nerve is transected and can be sutured without generating an undue tension [3]

  1. Page 2, second paragraph: the authors refer to prevcious studies that have been applying vein conduits for peripheral nerve repair, but they do not sufficiently report on the type of nerves that are commonly repaired with this kind of autologous graft. To the knowledge of this reviewer this is mainly digital nerves that present with short defect lengths and that are also often repaired with so-called muscle-in-vein grafts. The authors are asked to provide more specific information regarding the proven applicability of vein or muscle-in-vein conduits. Also it should be mentioned that to some extent similar limitations that are evident for autologous nerve grafts are also limiting the use of autologous vein grafts, e.g. in long gap repair, the availability of long venous grafts maybe considerably limited.

Response: The reviewer is right about the long gap repair. The vein may collapse and the insertion of a muscle within the conduit is beneficial. Although it is not relevant for the current study in which rats had to face a small loss of substance, we included a sentence about the muscle-in-vein grafts. The text now reads:

In addition, when the segment defect is long enough to possibly induce a collapse of the bridge, the inserted vein can be filled with fresh skeletal muscle (for a review, Battiston et al, 2007).

  1. Page 2, paragraph 4: The authors should also refer to literature already available for the use of olfactory ensheathing cells in peripheral nerve repair (e.g. Radtke, Kocsis Int J Mol Sci 2012 13(10) 12911-24, doi: 10.3390/ijms131012911, the review is summarizing some previous studies performed by expert groups – but more studies have been published more recently).

Response: As recommended by the reviewer, we inserted a sentence on the use of ensheathing cells for repairing peripheral nerve. It now reads:  

Besides, cell therapy can be performed as an additional approach. Human stem cells stand as a material of choice for scientists who wish to implement regenerative medicine [13]. However, to achieve these exciting promises, important choices have to be made, the first of which being the selection of the most appropriate stem cell subtype. Among the top candidates for brain research, we can cite the human nasal olfactory mucosa, home of a permanent neurogenesis [14], which harbors multipotent stem cells [15,16], belonging to the ecto-mesenchymal stem cell family [17], and also ensheathing cells that have been assessed for repairing peripheral nerves (for a review, Radtke, Kocsis Int J Mol Sci 2012, PMID 25765445).

  1. Page 2, paragraph 5: The summary of the study performed by the authors already uncovers the two main limitations. First of all, a 3mm nerve gap is very likely to regenerated spontaneously in the rat sciatic nerve or its branches such as the peroneal nerve investigated in this study. Second, a delay of two weeks does not allow for major changes to have occurred in the denervated distal nerve or its target organs – a minimal period of 6 weeks seems more appropriate to this reviewer, the authors are referred to existing literature, such as the work of Tessa Gordon and her colleagues and other contributions (e.g. Sulaiman OA, Gordon et al. Glia 2000, 32, 234-46 or Neurosurgery 2009,65:A 105-14; Stenberg et al. BMC Neuroscience 2017, 18:53). 10.1002/10981136(200012)32:3<234::aid-glia40>3.0.co;2-3

Response: We fully understand reviewer’s concerns. However, his/her remarks are appropriate when an acute repair strategy is considered. We wish to reiterate here that the focus of the article was primarily on a delayed repair of transected nerves. Such a clinical condition induces dramatic changes and, as demonstrated in the manuscript, the grafting of a vein, two weeks post-surgery, over a 3mm gap, is not sufficient to allow a satisfactory regeneration. Alternatively, if we had observed a comprehensive recovery in the DVG group, we would have had to repeat the experiment with the temporal and spatial gaps recommended by the reviewer.

Materials and Methods

  1. Section 2.2.: It is not sufficiently clear from this section, why the numbers of animals subjected to each experimental group display such a big variation – from 8 up to 22 animals per group. Also it is not described with enough clarity, when the stem cells have been injected, always two weeks after the initial repair and the repair was eventually delayed for two weeks? The later would have resulted in stem cell injection for the DVG-SC group at a time four weeks after the initial surgery?

Response: The reviewer rightly points out the large number (n=22) of rats included in the DVG-SC group. To tell the truth, this figure includes two sub-groups: 13 rats were used for the experiments described in the manuscript and 9 for the experiment assessing a putative migration of stem cells in other organs. To clarify this issue, the paragraph has been modified. It now reads:

Rats were randomized into five groups: 1) Control group (n=8) in which no surgery was performed, 2) Immediate Vein Graft (IVG) group (n=14) in which a segment of 3 mm peroneal nerve was removed and a vein conduit of 1 cm was immediately grafted between the two nerve stumps, 3) Immediate Vein Graft and Stem Cells (IVG-SC) group (n=13) in which a segment of 3 mm peroneal nerve was removed, a vein conduit of 1 cm was immediately grafted between the two nerve stumps and filled two weeks later with one million stem cells, 4) Delayed Vein Graft (DVG) group (n=10) in which a segment of 3 mm peroneal nerve was removed and a vein conduit of 1 cm was grafted, two weeks later, between the two nerve stumps, 5) Delayed Vein Graft and Stem Cells (DVG-SC) group (n=13) in which a segment of 3 mm peroneal nerve was removed and a vein conduit of 1 cm, filled with one million stem cells, was grafted, two weeks later, between the nerve stumps. In addition, 9 DVG-SC rats were included in order to assess the putative presence of stem cells in other organs.

On the second point (time of stem cell grafting), we agree that our wording may be misleading. For the IVG-SC group, stem cells were implanted two weeks after the vein insertion. For the DVG-SC group, cells were grafted at the time of vein insertion, e.g. at Week 2 post-surgery. The text has been modified accordingly and now reads:

Animals were deeply anesthetized (isoflurane 3%, Isoflurin®, Axience Santé Animale SAS, Pantin, France) and surgical procedures were performed aseptically under binoculars. For the IVG, IVG-SC, DVG and DVG-SC groups, the peroneal nerve from the left limb was dissected free from the surrounding tissues and cut on a 3 mm length, 5 mm away from the sciatic nerve trifurcation (i.e. starting point of tibial nerve, common peroneal nerve and caudal sural cutaneous nerve). A 1 cm long femoral vein was removed from the contralateral side of the nerve injury, washed in saline (NaCl 0.9%) and grafted either immediately (IVG and IVG-SC groups) or two weeks later (DVG and DVG-SC groups). The two nerve stumps were inserted into the vein, leaving a gap of 3 mm between the proximal and distal nerve stumps. The graft was fixed using three or four 9-0 monofilament non-absorbable sutures (Ethilon® 9-0, Ethicon Inc. Jonhson & Johnson, Sommerville, New Jersey, USA) for each stump as well as biological thrombin/fibrinogen glue (Tisseel, Baxter). For the IVG-SC and DVG-SC groups, one million of OEMSCs were resuspended in 5 l of thrombin, injected in the vein (IVG-SC and DVG-SC groups), using a Hamilton syringe (Hamilton Company, Bonaduz, Switzerland), filled with 5 l of fibrinogen. Stem cells were implanted two weeks after the vein insertion for the IVG-SC group and at the time of vein insertion for the DVG-SC group. In other words, in both cases, cells were grafted two weeks after the initial surgery. Muscles and skin were immediately stitched using non-absorbable monofilament sutures (Ethilon® 3-0, Ethicon Inc. Jonhson & Johnson, Sommerville, New Jersey, USA).

  1. The study design lacks an appropriate control group that should have been composed of autologous nerve grafting.

Response: We agree with the reviewer. The most appropriate control group would have included an autologous nerve graft. However, inserting a 3mm long nerve is almost impossible. Cutting properly such a small section and then interposed it on each side would have been too challenging. In addition, the nerve section would have been too damaged to provide reliable results.

  1. Section 2.3.: It is not clear, how the vein graft was treated in the period from initial harvest until implantation in the DVG and DVG-SC groups. Also it is not sufficiently described, how collapsing of the vein graft was prevented in IVG and DVG animals. Was the vehicle 5µl of each thrombin and fibrinogen also applied to these grafts? How was collapsing then prevented for the DVG-SC during the two weeks before SC injection? A collapse of the vein graft may prevent sufficient regeneration and therefore needs specific consideration.

Response: We are sorry for a misleading formulation. Whatever the group, the vein was always inserted immediately after collection. The text has been corrected accordingly:

A 1 cm long femoral vein was removed from the contralateral side of the nerve injury, washed in saline (NaCl 0.9%) and grafted immediately.

Furthermore, no collapse of the vein was ever observed, whatever the group considered. This is probably due to the fact the lost segment of nerve was small. Indeed, experimental studies have shown that the lumen of the vein conduit remains patent throughout the process of nerve regeneration for conduits up to 10 mm in length (Tseng et al., 2003).

  1. Section 2.3.: How was pain relieved post surgery?

Response: In line with reviewer’s request, we inserted a paragraph in the M&M section that now reads:

Clinical signs of pain or unpleasant sensations (self-mutilation, prostration, hyperactivity, anorexia and weight loss) were examined daily. All animals were treated with an injection of buprenorphine (0.5mg/kg subcutaneously) post-surgery. No other administration of this antalgic was performed, indicating that no sign of pain was observed.

  1. Section 2.6.: The authors have applied a quite comprehensive electrodiagnostic testing that is, unfortunately, considerably different from the commonly applied protocols (kindly refer to X. Navarro European Journal of Neuroscience, Vol. 43, pp. 271–286, 2016 doi:10.1111/ejn.13033). It is not clear to this reviewer what the benefits of the specific measurements are in comparison to commonly applied techniques. Also it is not clear, why the contralateral healthy nerve or muscle (section 2.7.) could not serve as individual control and why a sham group had to be installed.

Response: The electrodiagnostic testing used to evaluate the functional recovery was previously validated in several studies (Pertici et al., 2014; Chabas et al., 2013; Chabas et al., 2009; Alluin et al., 2009; Chabas et al., 2008; Alluin et al., 2006; Marqueste et al., 2004; Decherchi et al., 2001). These tests can be considered conventional, as those used by X. Navarro (2006). They allow the study of a part of the sensory field, namely the metabosensitive afferent fibers that are poorly considered in studies on peripheral nerve regeneration.

Furthermore, the controlateral healthy nerve or muscle cannot be used as control because, as previously demonstrated in several studies (Pertici et al., 2014; Moyne-Bressand et al., 2017…) measuring the weight-bearing distribution (DWB) or using electrophysiological tests (when one side is deficient/impeded), a compensation takes place on the healthy side, modifying the anatomo-functional properties of the latter. Using the healthy side as a control would distort the results. This is why we compared our results to those obtained in animals without lesion.

  1. Section 2.9.: It is not sufficiently clear what staining protocols were used. Was there a double-staining of myelin applied – detection of P0 protein AND Luxol fast blue staining? Further it is described that DAB was used for viszualising the staining which could have been detected in light microscopy as the luxol staininig, but the authors describe analysis with a TIRF microscope? How exactly were the numbers of axons, axon areas and myelin content determined in the sections? Commonly unbiased stereology and nerve morphometry are performed on tissue that has been post-fixed for myelin detection and embedded in epoxid and subsequently sectioned in ultra-thin nerve cross sections that are then available for light microscopy (e.g. Geuna S., J Comp Neurol 427:333-339).

Response: The two stainings were performed separately. As described, P0 was stained using immunohistochemistry whereas myelin was marked with Luxol fast blue. The text has been modified and now reads:

In addition, using alternate sections, myelin was also stained using the conventional Luxol method. Nerve sections were incubated overnight at 56°C in 0.1% Luxol (Clinisciences, Nanterre, France) solution (in 95% ethanol, 5% glacial acid acetic). After two washings in 95% ethanol and distilled water, sections were differentiated 30 sec first, in a 0.05% Lithium carbonate solution followed by another differentiation step in 70% ethanol for 30 sec. After washing in distilled water, sections were incubated in 95% ethanol for 5 min, in 100% ethanol twice for 5 min, and finally in Histo-Choice® (Sigma Aldrich-Merck) twice for 5 min.

About the microscope used for visualization, we are now more precise. The text now reads:

Stained sections were examined using an Apotome microscope (Apotome v2, AxioObserver Z1, Zeiss) which was associated with high-resolution camera (Caméra CMOS Orca flash 4.0 v2, Zeiss). The slides were digitized and analyzed with ImageJ (NIH) software. Axon numbers, axon areas and myelin content were measured in each group of animals.

Results

  1. Page 7, 3rd paragraph: Commonly muscle weight ratio is determined by taking the weight of the muscle from the lesioned and the healthy contralateral leg, why did the authors use a sham control?

Response: Please, see comment above (point 5. Section 2.6)

  1. Figure 1, B: How do the authors explain the worsening of the PFI performance in the DVG group at w12?

Response: There is no significant worsening of the PFI performance in the DVG group at W12. Statistical analysis does not indicate a statistically significant difference with scores measured at W10 and W8.

  1. Figure 2 and Figure 3, B/D: How do the authors explain that the DVG-SC group even displayed better recovery than the IVG-SC group – should the degeneration during the delay not have aggravated the conditions prior to repair surgery?

Response: We are surprised by this remark. As clearly mentioned, there is no significant difference between IVG and DVG, IVG-SC and DVG-SC groups.

  1. Section 3.5.: How the g-ratio was determined is not sufficiently described.

Response: We thank the reviewer for pointing out this missing information. It has been corrected and the text now reads:

To assess G-ratio (i.e. the ratio between the diameter of the axon and the outer diameter of the myelinated fibre), slides were coded, 5 regions of interest in each section were randomly chosen and data analysis was performed blindly.

  1. Representative photomicrographs should be provided for illustrating the histology results.

Response: An additional figure, composed of representative picture of all groups, is now included in the manuscript as supplementary figure 2.

  1. The authors should comment on the distally increased number of axons they detected in their IVG-SC and DVG-SC groups. Increased sprouting could be valuable at mid-graft level but should not occur pre-dominantly distal to the graft, where axons could form mini-fasicles and neuroma in continuity, the latter could represent a secondary impairment to functional recovery.

Response: We are not too sure to understand the reviewer’s comment. Statistically speaking, there is no significant increase in the number of axons in the distal part, when compared to the central and proximal parts, although the graph indicates a trend in the IVG-SC group.

  1. Figure 4: Usually axonal counts should be reported as nerve fiber density (number of axons/ square mm) this way also different cross-sectional areas are appropriately considered. Where the cross-sectional areas not differing between the groups and from proximal to distal along the nerve and the grafts?

Response: We didn’t report the ratio nb axons/mm2 because no inter-group difference was observed. We inserted a sentence in the text which now reads:

The number of axons/mm2 was also calculated. In absence of a significant difference with the results reported above, they are not displayed.

Discussion

  1. Section 4.1.: The authors should again more precise in their report of “good healing”. What is the functional outcome to be expected for what type of graft and what kind of nerve to be repaired with the same.

Response: In accordance with reviewer’s suggestion, we inserted a sentence with an attached reference. It now reads:

The most appropriate functional outcome to be measured, according to the animal species, the type of injured nerve and the nature of the experiment (pre-clinical versus clinical), can be found in a recent review (Ronchi et al, 2019).

  1. The authors should discuss their findings in the context of the clinical use and pre-clinical evaluation of muscle-in-vein grafts.

Response: Please see comment above and sentence added in 4.1.

  1. Section 4.2.: The conditions to be expected for nerves, Schwann cells, and motor endplates after chronic denervation should be discussed in more detail and also in consideration of the tremendous contribution Tessa Gordon and colleagues have provided to our knowledge.

Response: We understand reviewer’s request. However, we find difficult to insert a full paragraph without disrupting the flow of the discussion. Instead, we inserted a review of Tessa Gordon that summarizes the benefit of an electrical stimulation on nerve regeneration. The added sentence now reads:

To speed the process of regeneration, it could be envisioned to use electrical stimulation, a technique that has been successfully used in animal models and humans, grafted with autologous nerve segments (for a review, Gordon, 2016)

  1. Page 12, 7th paragraph: There are more reports to be considered with regard to recent publications of similar approaches, e.g. Gu J et al., Neural Regen Res 2019 14 124-31.

Response: As recommended by the reviewer, we inserted a sentence on the successful use of olfactory ensheathing for the repair of facial nerves. It now reads:

Of note, an olfactory bulb-associated cell type, namely the olfactory ensheathing cells, promotes facial regeneration and functional recovery when implanted in a rat transected facial nerve (Gu et al, 2018).

  1. Page 13, 1st paragraph: there is more information available on the exosome functions, the authors should consider for their discussion also: Xia B et al. Front Cell Neurosci 13:548.

Response: As recommended by the reviewer, we inserted the above-mentioned reference. The text now reads:

To test this hypothesis, our study could have included additional experimental groups in which the vein would have been filled with extracellular vesicles released from stem cells, as demonstrated by a team using exosomes from olfactory ensheathing cells (Xia et al, 2019).

Minor points

  1. Page 6, 3rd paragraph: does “medial” nerve section stand for what has been described as “middle part” in the first line of the same page?

Response: This discrepancy has been corrected.

  1. Legend of Figure 2: “…a significant increase in the …”

Response: This misspelling has been corrected.

  1. Page 10, 2nd paragraph: “For the DVG animals, compared to the Control …”

Response: The missing letter has been added.

  1. Page 12, 6th paragraph: “neurites” should be replaced by “axons”

Response: As recommended, the word neurites has been replaced by axons.

Round 2

Reviewer 2 Report

The authors have provided some minor revisions for their manuscript which, unfortunately, do not meet the expectations of this reviewer for appropriate improvement of the manuscript.

Critical points are listed below:

1) Original reviewer comment: Page 2, second paragraph: the authors refer to previous studies that have been applying vein conduits for peripheral nerve repair, but they do not sufficiently report on the type of nerves that are commonly repaired with this kind of autologous graft. To the knowledge of this reviewer this is mainly digital nerves that present with short defect lengths and that are also often repaired with so-called muscle-in-vein grafts. The authors are asked to provide more specific information regarding the proven applicability of vein or muscle-in-vein conduits. Also it should be mentioned that to some extent similar limitations that are evident for autologous nerve grafts are also limiting the use of autologous vein grafts, e.g. in long gap repair, the availability of long venous grafts maybe considerably limited.

Author Response: The reviewer is right about the long gap repair. The vein may collapse and the insertion of a muscle within the conduit is beneficial. Although it is not relevant for the current study in which rats had to face a small loss of substance, we included a sentence about the muscle-in-vein grafts. The text now reads: In addition, when the segment defect is long enough to possibly induce a collapse of the bridge, the inserted vein can be filled with fresh skeletal muscle (for a review, Battiston et al, 2007).

New comment: The introduction still points out the significance of nerve gaps and the limitations existing for repairing extended defects, it does still not provide a differentiated view on what vein grafts could offer and what limitations exist also for them. Either the authors would have needed to indicate already in their introduction that these scenarios are not relevant for their study (so would have to omit the related text) or they should have responded to my comment in a more careful way.

2) Original reviewer comment: Page 2, paragraph 4: The authors should also refer to literature already available for the use of olfactory ensheathing cells in peripheral nerve repair (e.g. Radtke, Kocsis Int J Mol Sci 2012 13(10) 12911-24, doi: 10.3390/ijms131012911, the review is summarizing some previous studies performed by expert groups – but more studies have been published more recently).

Author Response: As recommended by the reviewer, we inserted a sentence on the use of ensheathing cells for repairing peripheral nerve. It now reads:  Besides, cell therapy can be performed as an additional approach. Human stem cells stand as a material of choice for scientists who wish to implement regenerative medicine [13]. However, to achieve these exciting promises, important choices have to be made, the first of which being the selection of the most appropriate stem cell subtype. Among the top candidates for brain research, we can cite the human nasal olfactory mucosa, home of a permanent neurogenesis [14], which harbors multipotent stem cells [15,16], belonging to the ecto-mesenchymal stem cell family [17], and also ensheathing cells that have been assessed for repairing peripheral nerves (for a review, Radtke, Kocsis Int J Mol Sci 2012, PMID 25765445).

New comment: The respective paragraph is still mostly focused on applications of olfactory mucosa cells for central nervous system repair and not well differentiating between olfactory stem cells and olfactory ensheathing cells and also negating recent literature on the use of olfactory stem cells for repairing peripheral nerves (e.g. Roche P et al., Stem Cells Transl Med 2017, doi: 10.1002/sctm.16-0420). Therefore, just including an extension to the paragraph by citing an older review article, that was simply meant to guide the authors’ attention, is not an appropriate response to my request. The authors should have been demonstrating more willingness to include previous work that has been done with this kind of cells in models of peripheral nerve injury and repair, since it seems to be highly relevant for their study.

3) Original reviewer comment:  Page 2, paragraph 5: The summary of the study performed by the authors already uncovers the two main limitations. First of all, a 3mm nerve gap is very likely to regenerated spontaneously in the rat sciatic nerve or its branches such as the peroneal nerve investigated in this study. Second, a delay of two weeks does not allow for major changes to have occurred in the denervated distal nerve or its target organs – a minimal period of 6 weeks seems more appropriate to this reviewer, the authors are referred to existing literature, such as the work of Tessa Gordon and her colleagues and other contributions (e.g. Sulaiman OA, Gordon et al. Glia 2000, 32, 234-46 or Neurosurgery 2009,65:A 105-14; Stenberg et al. BMC Neuroscience 2017, 18:53). 10.1002/10981136(200012)32:3<234::aid-glia40>3.0.co;2-3

Author Response: We fully understand reviewer’s concerns. However, his/her remarks are appropriate when an acute repair strategy is considered. We wish to reiterate here that the focus of the article was primarily on a delayed repair of transected nerves. Such a clinical condition induces dramatic changes and, as demonstrated in the manuscript, the grafting of a vein, two weeks post-surgery, over a 3mm gap, is not sufficient to allow a satisfactory regeneration. Alternatively, if we had observed a comprehensive recovery in the DVG group, we would have had to repeat the experiment with the temporal and spatial gaps recommended by the reviewer.

New comment: I’m afraid the authors did not understand my concerns. It is indeed the short period of delay that I’m wondering about. I would have liked to see arguments that already after this period there is a significant change in the ability of the nerves to regenerate. Most of the studies analyzing delayed repair approaches use longer periods and no clear explanation is given by the authors why they did decide for only 2 weeks of delay.

4) Original reviewer comment:  The study design lacks an appropriate control group that should have been composed of autologous nerve grafting.

Author Response: We agree with the reviewer. The most appropriate control group would have included an autologous nerve graft. However, inserting a 3mm long nerve is almost impossible. Cutting properly such a small section and then interposed it on each side would have been too challenging. In addition, the nerve section would have been too damaged to provide reliable results.

New comment: The argument for not including an autograft repair group is not at all convincing. If dissecting a 3mm piece of the nerve was so complicated for the experimenters, why did they not choose a  larger gap right away? Probably that would have also brought up more convincing results in the end.  Also, if the dissection of a 3mm segment by the experimenter did result in largely damaged transection sites, how did this impact the regeneration capacity in the other groups? Was this the reason why the IVG group did not demonstrate already a very good (spontaneous) recovery? The study design should have had been adapted to the microsurgical capacities of the experimenter.

5) Original reviewer comment:  … Also it is not sufficiently described, how collapsing of the vein graft was prevented in IVG and DVG animals. Was the vehicle 5µl of each thrombin and fibrinogen also applied to these grafts? How was collapsing then prevented for the DVG-SC during the two weeks before SC injection? A collapse of the vein graft may prevent sufficient regeneration and therefore needs specific consideration.

Author Response: … Furthermore, no collapse of the vein was ever observed, whatever the group considered. This is probably due to the fact the lost segment of nerve was small. Indeed, experimental studies have shown that the lumen of the vein conduit remains patent throughout the process of nerve regeneration for conduits up to 10 mm in length (Tseng et al., 2003).

New comment: Thank you. Tseng et al. 2003 did investigate a 10 mm nerve gap and showed that regrowing axons have bridged it by day 19. This brings up an additional concern: For the immediate repair the experimenters accepted a good possibility that the regeneration process did already start after just putting a vein graft, than they manipulated  the lesion and repair side, again not knowing if they eventually induced a second lesion, e.g. by compressing the freshly grown axons in the graft during injection of the thrombin/ fibrinogen/ cell clot two weeks later again  - this may have biased the outcome of the study by putting a disadvantage to the IVG-SC group.

Also, the response of the authors clearly indicates that stem cells have been applied to the vein grafts using a vehicle but in their control condition no vehicle was injected. This does not seem appropriate.  

6) Original reviewer comment:  Section 2.6.: The authors have applied a quite comprehensive electrodiagnostic testing that is, unfortunately, considerably different from the commonly applied protocols (kindly refer to X. Navarro European Journal of Neuroscience, Vol. 43, pp. 271–286, 2016 doi:10.1111/ejn.13033). It is not clear to this reviewer what the benefits of the specific measurements are in comparison to commonly applied techniques. Also it is not clear, why the contralateral healthy nerve or muscle (section 2.7.) could not serve as individual control and why a sham group had to be installed.

Author Response:  The electrodiagnostic testing used to evaluate the functional recovery was previously validated in several studies (Pertici et al., 2014; Chabas et al., 2013; Chabas et al., 2009; Alluin et al., 2009; Chabas et al., 2008; Alluin et al., 2006; Marqueste et al., 2004; Decherchi et al., 2001). These tests can be considered conventional, as those used by X. Navarro (2006). They allow the study of a part of the sensory field, namely the metabosensitive afferent fibers that are poorly considered in studies on peripheral nerve regeneration.

Furthermore, the controlateral healthy nerve or muscle cannot be used as control because, as previously demonstrated in several studies (Pertici et al., 2014; Moyne-Bressand et al., 2017…) measuring the weight-bearing distribution (DWB) or using electrophysiological tests (when one side is deficient/impeded), a compensation takes place on the healthy side, modifying the anatomo-functional properties of the latter. Using the healthy side as a control would distort the results. This is why we compared our results to those obtained in animals without lesion.

New comment: The authors were obviously not willing to introduce the techniques they used in a more detailed way to the reader, this, however, would have been an appropriate way to respond to my question. They also did not obviously not acknowledge that X. Navarro European Journal of Neuroscience, Vol. 43, pp. 271–286, 2016 doi:10.1111/ejn.13033 is a review article written by a world leading expert in the field.

Furthermore, I’m still not convinced that lives of the sham animals could not have been made use of in a better way.

7) Original reviewer comment:  Section 2.9.: … How exactly were the numbers of axons, axon areas and myelin content determined in the sections? Commonly unbiased stereology and nerve morphometry are performed on tissue that has been post-fixed for myelin detection and embedded in epoxid and subsequently sectioned in ultra-thin nerve cross sections that are then available for light microscopy (e.g. Geuna S., J Comp Neurol 427:333-339).

Author Response:  … The text now reads: … The slides were digitized and analyzed with ImageJ (NIH) software. Axon numbers, axon areas and myelin content were measured in each group of animals.

New comment: This is not an appropriate response to my concern. The authors do still not provide evidence that they have used unbiased quantification methods similar to those accepted as the state-of-the-art.

8) Original reviewer comment: Page 7, 3rd paragraph: Commonly muscle weight ratio is determined by taking the weight of the muscle from the lesioned and the healthy contralateral leg, why did the authors use a sham control?

Author Response:  Please, see comment above (point 5. Section 2.6) – new point 5

New comment: In this case, however, the authors should have provided data on to which degree the sham control and experimental animals were matching in body weight, scale of the lower limbs etc.  

9) Original reviewer comment: Figure 1, B: How do the authors explain the worsening of the PFI performance in the DVG group at w12?

…. Figure 2 and Figure 3, B/D: How do the authors explain that the DVG-SC group even displayed better recovery than the IVG-SC group – should the degeneration during the delay not have aggravated the conditions prior to repair surgery?

Author Response: There is no significant worsening of the PFI performance in the DVG group at W12. Statistical analysis does not indicate a statistically significant difference with scores measured at W10 and W8.

…..We are surprised by this remark. As clearly mentioned, there is no significant difference between IVG and DVG, IVG-SC and DVG-SC groups.

New comment: Still the shown fluctuation and also somehow conflicting data resulting from the different tests could have been commented more carefully in the manuscript or if there was really no conflict the text could have been improved for clarity.

10) Original reviewer comment: Section 3.5.: How the g-ratio was determined is not sufficiently described.

Author Response: We thank the reviewer for pointing out this missing information. It has been corrected and the text now reads: To assess G-ratio (i.e. the ratio between the diameter of the axon and the outer diameter of the myelinated fibre), slides were coded, 5 regions of interest in each section were randomly chosen and data analysis was performed blindly.

New comment: This is unfortunately again not an appropriate response to my concern. The authors do still not provide sufficient detail  - what magnification was used, how many axons per animal were analysed, how was random selection of these axons guaranteed?

11) Original reviewer comment: Representative photomicrographs should be provided for illustrating the histology results.

Author Response: An additional figure, composed of representative picture of all groups, is now included in the manuscript as supplementary figure 2.

 New comment: The new figure is of incredible low quality, it does not provide sufficient insight in how the analysis of axonal regeneration was performed. But it demonstrates that the quality of the analysed section was of high variability and also the nerve cross-sectional areas were considerably differing.

12) Original reviewer comment: Figure 4: Usually axonal counts should be reported as nerve fiber density (number of axons/ square mm) this way also different cross-sectional areas are appropriately considered. Where the cross-sectional areas not differing between the groups and from proximal to distal along the nerve and the grafts?

Author Response: We didn’t report the ratio nb axons/mm2 because no inter-group difference was observed. We inserted a sentence in the text which now reads: The number of axons/mm2 was also calculated. In absence of a significant difference with the results reported above, they are not displayed.

New comment: This response is surprising. As mentioned above, from the new supplementary figure 2, it is obvious that the nerve cross-sectional areas were differing between the samples and most likely also the groups. But what is really surprising is that the authors decided not to show values for the nerve fiber density because this parameter did not demonstrate inter-group differences. Instead they decided to show axonal numbers and claim in their manuscript text that the differences shown for these values were the same for the analysis of nerve fiber density. This does not seem appropriate to me.

13) Original reviewer comment: Section 4.1.: The authors should again more precise in their report of “good healing”. What is the functional outcome to be expected for what type of graft and what kind of nerve to be repaired with the same.

…The authors should discuss their findings in the context of the clinical use and pre-clinical evaluation of muscle-in-vein grafts.

Author Response:  In accordance with reviewer’s suggestion, we inserted a sentence with an attached reference. It now reads: The most appropriate functional outcome to be measured, according to the animal species, the type of injured nerve and the nature of the experiment (pre-clinical versus clinical), can be found in a recent review (Ronchi et al, 2019).

… Please see comment above and sentence added in 4.1.

New comment: The reference inserted is referring to the rat median nerve model that does not relate at all the study presented by the authors. It would have been more informative to add information on regeneration parameters studied by others. Only referencing review articles does not seem appropriate.

Furthermore, the inserted reference gives no indication on how the authors would rank their study outcome into the context of the clinical use and pre-clinical evaluation of muscle-in-vein grafts.

14) Original reviewer comment: Section 4.2.: The conditions to be expected for nerves, Schwann cells, and motor endplates after chronic denervation should be discussed in more detail and also in consideration of the tremendous contribution Tessa Gordon and colleagues have provided to our knowledge.

Author Response:  We understand reviewer’s request. However, we find difficult to insert a full paragraph without disrupting the flow of the discussion. Instead, we inserted a review of Tessa Gordon that summarizes the benefit of an electrical stimulation on nerve regeneration. The added sentence now reads: To speed the process of regeneration, it could be envisioned to use electrical stimulation, a technique that has been successfully used in animal models and humans, grafted with autologous nerve segments (for a review, Gordon, 2016)

New comment: In deed, adding a paragraph on the consequences of delayed repair and whaich periods can be defined here would have been an appropriate action. The paragraph could have also been added to the introduction. The added sentence, however, appears like an un-linked annex and has no added value. Also the topic of electrical stimulation of the proximal nerve ending during the reconstruction surgery is not at all linked to the study the authors presented here.

15) New comment: The authors further added other references as annexes to their previous text without linking them appropriately to the context.  I was just referring to these publications as examples, the authors should have tried to read a little more around them and to find a more appropriate way or even better references to revise their manuscript in response to my concerns and for providing a more critical discussion.  

Author Response

Response to reviewer 

1) Original reviewer comment: Page 2, second paragraph: the authors refer to previous studies that have been applying vein conduits for peripheral nerve repair, but they do not sufficiently report on the type of nerves that are commonly repaired with this kind of autologous graft. To the knowledge of this reviewer this is mainly digital nerves that present with short defect lengths and that are also often repaired with so-called muscle-in-vein grafts. The authors are asked to provide more specific information regarding the proven applicability of vein or muscle-in-vein conduits. Also it should be mentioned that to some extent similar limitations that are evident for autologous nerve grafts are also limiting the use of autologous vein grafts, e.g. in long gap repair, the availability of long venous grafts maybe considerably limited.

Author Response: The reviewer is right about the long gap repair. The vein may collapse and the insertion of a muscle within the conduit is beneficial. Although it is not relevant for the current study in which rats had to face a small loss of substance, we included a sentence about the muscle-in-vein grafts. The text now reads: In addition, when the segment defect is long enough to possibly induce a collapse of the bridge, the inserted vein can be filled with fresh skeletal muscle (for a review, Battiston et al, 2007).

New comment: The introduction still points out the significance of nerve gaps and the limitations existing for repairing extended defects, it does still not provide a differentiated view on what vein grafts could offer and what limitations exist also for them. Either the authors would have needed to indicate already in their introduction that these scenarios are not relevant for their study (so would have to omit the related text) or they should have responded to my comment in a more careful way. 

New author response: Somehow, the reviewer asks us to write comparison of nerve vs. vein vs. muscle in vein. This is an interesting exercise that could be the main topic of a review. However, we feel that it’s too long to be inserted in the introduction of a research paper. A referenced sentence has been added. It now reads: Overall, a venous conduit is a suitable clinical material since it is biologically inert, inexpensive, biocompatible, thin, flexible and transparent.

2) Original reviewer comment: Page 2, paragraph 4: The authors should also refer to literature already available for the use of olfactory ensheathing cells in peripheral nerve repair (e.g. Radtke, Kocsis Int J Mol Sci 2012 13(10) 12911-24, doi: 10.3390/ijms131012911, the review is summarizing some previous studies performed by expert groups – but more studies have been published more recently).

Author Response: As recommended by the reviewer, we inserted a sentence on the use of ensheathing cells for repairing peripheral nerve. It now reads:  Besides, cell therapy can be performed as an additional approach. Human stem cells stand as a material of choice for scientists who wish to implement regenerative medicine [13]. However, to achieve these exciting promises, important choices have to be made, the first of which being the selection of the most appropriate stem cell subtype. Among the top candidates for brain research, we can cite the human nasal olfactory mucosa, home of a permanent neurogenesis [14], which harbors multipotent stem cells [15,16], belonging to the ecto-mesenchymal stem cell family [17], and also ensheathing cells that have been assessed for repairing peripheral nerves (for a review, Radtke, Kocsis Int J Mol Sci 2012, PMID 25765445).

New comment: The respective paragraph is still mostly focused on applications of olfactory mucosa cells for central nervous system repair and not well differentiating between olfactory stem cells and olfactory ensheathing cells and also negating recent literature on the use of olfactory stem cells for repairing peripheral nerves (e.g. Roche P et al., Stem Cells Transl Med 2017, doi: 10.1002/sctm.16-0420). Therefore, just including an extension to the paragraph by citing an older review article, that was simply meant to guide the authors’ attention, is not an appropriate response to my request. The authors should have been demonstrating more willingness to include previous work that has been done with this kind of cells in models of peripheral nerve injury and repair, since it seems to be highly relevant for their study.

New author response: We feel ashamed for having missed to cite this important study and we would have been grateful if the reviewer had mentioned this omission in his first series of comments. This article is now mentioned in the introduction and discussion chapters.

3) Original reviewer comment:  Page 2, paragraph 5: The summary of the study performed by the authors already uncovers the two main limitations. First of all, a 3mm nerve gap is very likely to regenerated spontaneously in the rat sciatic nerve or its branches such as the peroneal nerve investigated in this study. Second, a delay of two weeks does not allow for major changes to have occurred in the denervated distal nerve or its target organs – a minimal period of 6 weeks seems more appropriate to this reviewer, the authors are referred to existing literature, such as the work of Tessa Gordon and her colleagues and other contributions (e.g. Sulaiman OA, Gordon et al. Glia 2000, 32, 234-46 or Neurosurgery 2009,65:A 105-14; Stenberg et al. BMC Neuroscience 2017, 18:53). 10.1002/10981136(200012)32:3<234::aid-glia40>3.0.co;2-3

Author Response: We fully understand reviewer’s concerns. However, his/her remarks are appropriate when an acute repair strategy is considered. We wish to reiterate here that the focus of the article was primarily on a delayed repair of transected nerves. Such a clinical condition induces dramatic changes and, as demonstrated in the manuscript, the grafting of a vein, two weeks post-surgery, over a 3mm gap, is not sufficient to allow a satisfactory regeneration. Alternatively, if we had observed a comprehensive recovery in the DVG group, we would have had to repeat the experiment with the temporal and spatial gaps recommended by the reviewer.

New comment: I’m afraid the authors did not understand my concerns. It is indeed the short period of delay that I’m wondering about. I would have liked to see arguments that already after this period there is a significant change in the ability of the nerves to regenerate. Most of the studies analyzing delayed repair approaches use longer periods and no clear explanation is given by the authors why they did decide for only 2 weeks of delay.

New author response: On the one hand, the length of the delay was chosen to comply with the desire of clinicians to intervene as soon as possible and the need to allow sufficient time to purify several million stem cells. In a parallel study, performed at the hospital in GMP conditions, we observed that it takes 10 to 15 days to culture 10 million olfactory stem cells. On the other hand, the reviewer indicates that the time frame is too short to observe major deficits. We would like to reiterate one of our major observations: the vein alone, inserted two weeks after the initial trauma, does not allow the injured animals to regain normal locomotion. The text now reads: The chronic model mirrors the delayed repair associated to neglected wounds in human care and, in line with a clinical trial based on autologous stem cell transplantation, the time required for cultivating, in GMP conditions, about 10 millionsof human OEMSCs.

4) Original reviewer comment:  The study design lacks an appropriate control group that should have been composed of autologous nerve grafting.

Author Response: We agree with the reviewer. The most appropriate control group would have included an autologous nerve graft. However, inserting a 3mm long nerve is almost impossible. Cutting properly such a small section and then interposed it on each side would have been too challenging. In addition, the nerve section would have been too damaged to provide reliable results.

New comment: The argument for not including an autograft repair group is not at all convincing. If dissecting a 3mm piece of the nerve was so complicated for the experimenters, why did they not choose a larger gap right away? Probably that would have also brought up more convincing results in the end.  Also, if the dissection of a 3mm segment by the experimenter did result in largely damaged transection sites, how did this impact the regeneration capacity in the other groups? Was this the reason why the IVG group did not demonstrate already a very good (spontaneous) recovery? The study design should have had been adapted to the microsurgical capacities of the experimenter.

New author response: The comment of the reviewer on the microsurgical capacities of the experimenter is slightly offending. Surgeries were performed by experienced clinicians, experts in plastic and reconstructive surgery.

5) Original reviewer comment:  … Also it is not sufficiently described, how collapsing of the vein graft was prevented in IVG and DVG animals. Was the vehicle 5µl of each thrombin and fibrinogen also applied to these grafts? How was collapsing then prevented for the DVG-SC during the two weeks before SC injection? A collapse of the vein graft may prevent sufficient regeneration and therefore needs specific consideration.

Author Response: … Furthermore, no collapse of the vein was ever observed, whatever the group considered. This is probably due to the fact the lost segment of nerve was small. Indeed, experimental studies have shown that the lumen of the vein conduit remains patent throughout the process of nerve regeneration for conduits up to 10 mm in length (Tseng et al., 2003).

New comment: Thank you. Tseng et al. 2003 did investigate a 10 mm nerve gap and showed that regrowing axons have bridged it by day 19. This brings up an additional concern: For the immediate repair the experimenters accepted a good possibility that the regeneration process did already start after just putting a vein graft, than they manipulated  the lesion and repair side, again not knowing if they eventually induced a second lesion, e.g. by compressing the freshly grown axons in the graft during injection of the thrombin/ fibrinogen/ cell clot two weeks later again  - this may have biased the outcome of the study by putting a disadvantage to the IVG-SC group.

New author response: The reviewer may be right. Possibly, the injection of the cocktail stem cells/thrombin/fibrinogen temporarily altered the process of nerve regeneration. However, our data on locomotor recovery and axon regrowth indicate that the IVG-SC group displays improved outcomes when compared to the IVG group. In the future, as mentioned in the discussion, we aim to insert extracellular vesicles instead of stem cells. This may limit the consequence of a cell transplantation within a vein inserted two weeks earlier.

Also, the response of the authors clearly indicates that stem cells have been applied to the vein grafts using a vehicle but in their control condition no vehicle was injected. This does not seem appropriate.

New author Response: We are sorry for not having been more precise in our protocol. A 10 microliters solution of thrombin/fibrinogen was inserted in the vein of all IVG and DVG animals. This point is now clearly mentioned in the paragraph which now reads:  Rats were randomized into five groups: 1) Control group (n=8) in which no surgery was performed, 2) Immediate Vein Graft (IVG) group (n=14) in which a segment of 3 mm peroneal nerve was removed and a vein conduit of 1 cm was immediately grafted between the two nerve stumps and filled with 10 ml of thrombin/fibrinogen , 3) Immediate Vein Graft and Stem Cells (IVG-SC) group (n=13) in which a segment of 3 mm peroneal nerve was removed, a vein conduit of 1 cm was immediately grafted between the two nerve stumps and filled two weeks later with one million stem cells resuspended in 10ml of thrombin/fibrinogen, 4) Delayed Vein Graft (DVG) group (n=10) in which a segment of 3 mm peroneal nerve was removed and a vein conduit of 1 cm was grafted, two weeks later, between the two nerve stumps, and filled with 10 ul of thrombin/fibrinogen, 5) Delayed Vein Graft and Stem Cells (DVG-SC) group (n=13) in which a segment of 3 mm peroneal nerve was removed and a vein conduit of 1 cm, filled with one million stem cells, resuspended in 10 ml of thrombin/fibrinogen ,was grafted, two weeks later, between the nerve stumps. In addition, 9 DVG-SC rats were included in order to assess the putative presence of stem cells in other organs.

6) Original reviewer comment:  Section 2.6.: The authors have applied a quite comprehensive electrodiagnostic testing that is, unfortunately, considerably different from the commonly applied protocols (kindly refer to X. Navarro European Journal of Neuroscience, Vol. 43, pp. 271–286, 2016 doi:10.1111/ejn.13033). It is not clear to this reviewer what the benefits of the specific measurements are in comparison to commonly applied techniques. Also it is not clear, why the contralateral healthy nerve or muscle (section 2.7.) could not serve as individual control and why a sham group had to be installed.

 Author Response:  The electrodiagnostic testing used to evaluate the functional recovery was previously validated in several studies (Pertici et al., 2014; Chabas et al., 2013; Chabas et al., 2009; Alluin et al., 2009; Chabas et al., 2008; Alluin et al., 2006; Marqueste et al., 2004; Decherchi et al., 2001). These tests can be considered conventional, as those used by X. Navarro (2006). They allow the study of a part of the sensory field, namely the metabosensitive afferent fibers that are poorly considered in studies on peripheral nerve regeneration.

Furthermore, the controlateral healthy nerve or muscle cannot be used as control because, as previously demonstrated in several studies (Pertici et al., 2014; Moyne-Bressand et al., 2017…) measuring the weight-bearing distribution (DWB) or using electrophysiological tests (when one side is deficient/impeded), a compensation takes place on the healthy side, modifying the anatomo-functional properties of the latter. Using the healthy side as a control would distort the results. This is why we compared our results to those obtained in animals without lesion.

New comment: The authors were obviously not willing to introduce the techniques they used in a more detailed way to the reader, this, however, would have been an appropriate way to respond to my question. They also did not obviously not acknowledge that X. Navarro European Journal of Neuroscience, Vol. 43, pp. 271–286, 2016 doi:10.1111/ejn.13033 is a review article written by a world leading expert in the field.

Furthermore, I’m still not convinced that lives of the sham animals could not have been made use of in a better way.

New author response: All techniques performed in this article were previously used by world-renowned authors in the field of peripheral nerve lesions/repairs who published their work well before the review of Navarro et al. 2016. All these techniques were detailed in the articles cited in our manuscript. Moreover, the review of Navarro et al. is incomplete since it makes no reference to many previous works, particularly with regard to axonal regeneration of metabosensitive afferent pathways. For this reason, we cited original works related to our work rather than an incomplete review.

We are not too sure what are the SHAM animals evoked by the reviewer. A SHAM group could be a group of animals operated without lesion or a group of animals in which a nerve graft has been interposed between the two stumps. In the latter case, we do not see how this group could be compared to a group in which we inserted a venous segment. We avoided a SHAM group because i) the data from such a group would not have added anything more and ii) our study includes a Control group. The latter is sufficient to evaluate the effectiveness of our method of repair.

7) Original reviewer comment:  Section 2.9.: … How exactly were the numbers of axons, axon areas and myelin content determined in the sections? Commonly unbiased stereology and nerve morphometry are performed on tissue that has been post-fixed for myelin detection and embedded in epoxid and subsequently sectioned in ultra-thin nerve cross sections that are then available for light microscopy (e.g. Geuna S., J Comp Neurol 427:333-339).

Author Response:  … The text now reads: … The slides were digitized and analyzed with ImageJ (NIH) software. Axon numbers, axon areas and myelin content were measured in each group of animals.

New comment: This is not an appropriate response to my concern. The authors do still not provide evidence that they have used unbiased quantification methods similar to those accepted as the state-of-the-art.

New author response: We carefully read the publications on nerve regeneration by Geuna et al. All refer to the review mentioned by the reviewer. We read this review but could not find the details that would confirm that we are using the state-of-the-art protocol.

8) Original reviewer comment: Page 7, 3rd paragraph: Commonly muscle weight ratio is determined by taking the weight of the muscle from the lesioned and the healthy contralateral leg, why did the authors use a sham control?

Author Response:  Please, see comment above (point 5. Section 2.6) – new point 5

New comment: In this case, however, the authors should have provided data on to which degree the sham control and experimental animals were matching in body weight, scale of the lower limbs etc.

New author response: As explained above, a SHAM group would not have been a satisfactory addition. Furthermore, as mentioned in the first round of replies, when an animal's leg is injured, a compensation process takes place on the healthy side, modifying the anatomo-functional properties of the latter (Pertici et al., 2014; Moyne-Bressand et al., 2017...). For this reason, we set up a control group where no compensation was at work.

9) Original reviewer comment: Figure 1, B: How do the authors explain the worsening of the PFI performance in the DVG group at w12?

…. Figure 2 and Figure 3, B/D: How do the authors explain that the DVG-SC group even displayed better recovery than the IVG-SC group – should the degeneration during the delay not have aggravated the conditions prior to repair surgery?

Author Response: There is no significant worsening of the PFI performance in the DVG group at W12. Statistical analysis does not indicate a statistically significant difference with scores measured at W10 and W8.

…..We are surprised by this remark. As clearly mentioned, there is no significant difference between IVG and DVG, IVG-SC and DVG-SC groups.

New comment: Still the shown fluctuation and also somehow conflicting data resulting from the different tests could have been commented more carefully in the manuscript or if there was really no conflict the text could have been improved for clarity. 

New author response: It’s surprising to be asked to comment on trends. As scientists, we usually refrain to do such things and focus on what is statistically significant.

10) Original reviewer comment: Section 3.5.: How the g-ratio was determined is not sufficiently described.

Author Response: We thank the reviewer for pointing out this missing information. It has been corrected and the text now reads: To assess G-ratio (i.e. the ratio between the diameter of the axon and the outer diameter of the myelinated fibre), slides were coded, 5 regions of interest in each section were randomly chosen and data analysis was performed blindly.

New comment: This is unfortunately again not an appropriate response to my concern. The authors do still not provide sufficient detail  - what magnification was used, how many axons per animal were analysed, how was random selection of these axons guaranteed? 

New author response: Further details were added to the M&M section. It now reads: To assess G-ratio (i.e. the ratio between the diameter of the axon and the outer diameter of the myelinated fibre), slides were coded, 5 regions of interest in each section were randomly chosen and data analysis (100 fibers per group) was performed blindly at x100 magnification.

11) Original reviewer comment: Representative photomicrographs should be provided for illustrating the histology results.

Author Response: An additional figure, composed of representative picture of all groups, is now included in the manuscript as supplementary figure 2.

New comment: The new figure is of incredible low quality, it does not provide sufficient insight in how the analysis of axonal regeneration was performed. But it demonstrates that the quality of the analysed section was of high variability and also the nerve cross-sectional areas were considerably differing. 

New author response: We set up this additional figure in order to respond to reviewer’s request. As recommended by the reviewer, we selected low magnification pictures. We feel that it would have been more illustrating to display high magnification views. However, the very short delay for revising the manuscript (plus the Covid-19-associated difficulties to reach the laboratory) prevents us from modifying this figure. We propose to erase this figure.

12) Original reviewer comment: Figure 4: Usually axonal counts should be reported as nerve fiber density (number of axons/ square mm) this way also different cross-sectional areas are appropriately considered. Where the cross-sectional areas not differing between the groups and from proximal to distal along the nerve and the grafts?

Author Response: We didn’t report the ratio nb axons/mm2 because no inter-group difference was observed. We inserted a sentence in the text which now reads: The number of axons/mm2 was also calculated. In absence of a significant difference with the results reported above, they are not displayed.

New comment: This response is surprising. As mentioned above, from the new supplementary figure 2, it is obvious that the nerve cross-sectional areas were differing between the samples and most likely also the groups. But what is really surprising is that the authors decided not to show values for the nerve fiber density because this parameter did not demonstrate inter-group differences. Instead they decided to show axonal numbers and claim in their manuscript text that the differences shown for these values were the same for the analysis of nerve fiber density. This does not seem appropriate to me.

New author response: The data can be sent to the reviewer, if needed. We consider that non-significant results are of poor interest to the readers.

13) Original reviewer comment: Section 4.1.: The authors should again more precise in their report of “good healing”. What is the functional outcome to be expected for what type of graft and what kind of nerve to be repaired with the same.

…The authors should discuss their findings in the context of the clinical use and pre-clinical evaluation of muscle-in-vein grafts.

Author Response:  In accordance with reviewer’s suggestion, we inserted a sentence with an attached reference. It now reads: The most appropriate functional outcome to be measured, according to the animal species, the type of injured nerve and the nature of the experiment (pre-clinical versus clinical), can be found in a recent review (Ronchi et al, 2019).

… Please see comment above and sentence added in 4.1.

New comment: The reference inserted is referring to the rat median nerve model that does not relate at all the study presented by the authors. It would have been more informative to add information on regeneration parameters studied by others. Only referencing review articles does not seem appropriate.

New author response: To avoid a lengthy discussion on all potentially measured outcomes, we opted for this review. We feel it’s the appropriate response to reviewer’s comment.

Furthermore, the inserted reference gives no indication on how the authors would rank their study outcome into the context of the clinical use and pre-clinical evaluation of muscle-in-vein grafts.

New author response: Ranking our study among others that use various animal models, behavioural tests, electrophysiology recordings and histology techniques sounds an impossible exercise, not to mention that it can lead to immodesty.

14) Original reviewer comment: Section 4.2.: The conditions to be expected for nerves, Schwann cells, and motor endplates after chronic denervation should be discussed in more detail and also in consideration of the tremendous contribution Tessa Gordon and colleagues have provided to our knowledge.

Author Response:  We understand reviewer’s request. However, we find difficult to insert a full paragraph without disrupting the flow of the discussion. Instead, we inserted a review of Tessa Gordon that summarizes the benefit of an electrical stimulation on nerve regeneration. The added sentence now reads: To speed the process of regeneration, it could be envisioned to use electrical stimulation, a technique that has been successfully used in animal models and humans, grafted with autologous nerve segments (for a review, Gordon, 2016)

New comment: In deed, adding a paragraph on the consequences of delayed repair and whaich periods can be defined here would have been an appropriate action. The paragraph could have also been added to the introduction. The added sentence, however, appears like an un-linked annex and has no added value. Also the topic of electrical stimulation of the proximal nerve ending during the reconstruction surgery is not at all linked to the study the authors presented here.

New author response: In accordance with reviewer’s recommendation, we can erase the added sentence.

15) New comment: The authors further added other references as annexes to their previous text without linking them appropriately to the context.  I was just referring to these publications as examples, the authors should have tried to read a little more around them and to find a more appropriate way or even better references to revise their manuscript in response to my concerns and for providing a more critical discussion.  

New author response: We thank the reviewer for all his/her suggestions. They helped us to improve our manuscript and widen our research perspectives.